# Fundamental Limits of Visual Autoregressive Transformers: Universal Approximation Abilities

Yifang Chen [1]   Xiaoyu Li [2]   Yingyu Liang [3 4]   Zhenmei Shi [4]   Zhao Song [5]

## Abstract

We investigate the fundamental limits of transformer-based foundation models, extending our analysis to include Visual Autoregressive (VAR) transformers. VAR represents a big step toward generating images using a novel, scalable, coarse-to-fine "next-scale prediction" framework. These models set a new quality bar, outperforming all previous methods, including Diffusion Transformers, while having state-of-the-art performance for image synthesis tasks. Our primary contributions establish that, for single-head VAR transformers with a single self-attention layer and single interpolation layer, the VAR Transformer is universal. From the statistical perspective, we prove that such simple VAR transformers are universal approximators for any word-to-image Lipschitz functions. Furthermore, we demonstrate that flow-based autoregressive transformers inherit similar approximation capabilities. Our results provide important design principles for effective and computationally efficient VAR Transformer strategies that can be used to extend their utility to more sophisticated VAR models in image generation and other related areas.

## 1. Introduction

Transformer architectures have gained many attentions recently, and it has reshaped the landscape of modern machine learning. Transformer-based models have shown the state-of-the-art performance in many tasks, such as natural language processing (e.g., GPT-o1 (OpenAI, 2024), Llama 3.3 (Llama Team, 2024; AI, 2024), and Claude 3.5 (Anthropic, 2024)), computer vision, and generative modeling.

[1]The University of Chicago [2]University of New South Wales [3]The University of Hong Kong [4]University of Wisconsin–Madison [5]University of California, Berkeley. Correspondence to: Zhao Song <magic.linuxkde@gmail.com>.

*Proceedings of the 42$^{nd}$ International Conference on Machine Learning*, Vancouver, Canada. PMLR 267, 2025. Copyright 2025 by the author(s).

The core component in the transformer architecture is the self-attention mechanism, which captures long-range dependencies in data (Vaswani et al., 2017). A transformer-variants model, called the Visual Auto-regressive Transformer (VAR) (Tian et al., 2024), utilizes the transformer architecture to structure image synthesis. VAR Transformer uses "next-scale prediction" method to produce high quality images much more efficiently and outperform many standard diffusion models (Song et al., 2021). The iterative generation process of VAR demonstrates powerful performance in large-scale visual tasks, but the previous work is solely empirical, where the theoretical understandings of why VAR Transformers architecture excel in the image synthesis tasks is missing.

In addition to the VAR transfromer, flow-based generative methods (e.g., real-valued non-volume preserving (Real-NVP) and Glow) have also garnered attention for their ability to generate high-fidelity samples in an invertible and tractable manner. Recent efforts have integrated autoregressive decompositions with flow-based designs, giving rise to Flow AutoRegressive (FlowAR) (Ren et al., 2024) architectures. These models aim to blend the interpretability and stability of normalizing flows with the powerful representation learning of autoregressive transformers, potentially yielding more robust and scalable training dynamics. Similar to the studies on VAR transformers, the previous work on FlowAR focus on the empirical results but missing the theoretical understanding of why FlowAR excels other architetures.

To provide the theoretical analysis of the transformer-based architecture, the universal approximation theory is popular in recent studies. Standard transformers are proved to be univeral sequence-to-sequence approximators, which means that transformer is capable of representing any continuous mapping between input and output sequences given sufficient width and depth (Yun et al., 2020; Kajitsuka & Sato, 2024). Therefore, given the lack of theoretical analysis of VAR transformer, it is natural to ask the question:

*Are* VAR *Transformers the universal approximators?*

In this work, we extend the unviersal approximation result from (Yun et al., 2020; Kim et al., 2022; Kajitsuka & Sato,

2024; Hu et al., 2024b; Liu et al., 2025; Hu et al., 2025a) to prove that the VAR Transformer and some other variants are universal approximators. Our contribution can be summarized as follows: The key contributions of this paper are as follows:

- We formally establish that single-layer, single-head VAR Transformers are universal function approximators for Lipschitz image-to-image mappings , extending universality results to the VAR framework.

- Our results elucidate the theoretical expressiveness of VAR Transformers, showing how their fundamental components, including self-attention and up-sampling, enable them to approximate arbitrary continuous transformations.

- We provide insights into the broader implications of our findings for generative modeling, particularly in computer vision, where efficient and expressive architectures are essential for high-quality image synthesis. Building on the universal approximation property established for VAR Transformers, we show that the guarantee naturally extends to other autoregressive models as well.

Our theoretical contributions address two central gaps in theoretical understandings of generative transformer architectures, and it provides more insights to design more flexible and efficient generative models. Furthermore, our results provide a deeper understanding of why VAR and some other variants perform effectively in practice, such as even with the most minimal set up (single layer and single head), VAR transformer is capable of approximating complex functions.

## 2. Related Work

**Universal Approximation of Transformer**  Transformer architectures are universal approximators, which means they have the capabilities to approximate arbitrary sequence-to-sequence functions under mild conditions. (Yun et al., 2020) shows that deep transformers with stacked self-attention and feedforward layers can approximate any continuous mapping. (Jiang & Li, 2023) uses Kolmogorov-Arnold representations to derive the same results. (Alberti et al., 2023) extend the universality results to non-standard attention mechanisms, and (Kajitsuka & Sato, 2024; Hu et al., 2024b) explores the universality results in single-layer and single head transformers. Overall, these results build a strong foundations for the theoretical understandings on the expressiveness of transformer-based architectures.

**Auto-regressive Image Generation Models**  In image generation, auto-regressive models play a pivotal role by decomposing high-dimension distributions into a product

of conditions. PixelCNN (Van den Oord et al., 2016) and PixelSNAIL (Chen et al., 2018) generate one pixel at a time, which can capture local dependencies with convolutional architectures. Recent work use discrete latent tokenization, which facilitate generation at high resolutions. For example (Esser et al., 2021) introduces VQ-GAN that utilizes vector quantization with a GPT decoder, and VQVAE-2 from (Razavi et al., 2019) and RQ-Transformer from (Lee et al., 2022) propose multi-scale hierachies to capture coarse and fine-level structure. Based on these previous work, (Tian et al., 2024) designs the Visual AutoRegressive (VAR) Transformer which implement coase-to-fine "next-scale prediction" mechanism and combine transformer decoding with pyramid-shaped token representations. VAR Transformer achieves state-of-the-art performance in autoregressive image sythesis.

**Diffusion Models.**  Diffusion models (Ho et al., 2020; Rombach et al., 2022) excel in generating high-resolution images by iteratively refining noise into coherent visuals. Prominent examples, such as DiT (Peebles & Xie, 2023) and U-ViT (Bao et al., 2023), leverage probabilistic frameworks to learn data distributions effectively. Recent progress in diffusion-based image generation has focused on enhancing sampling techniques and training efficiency (Song & Ermon, 2019; Song et al., 2021; Lu et al., 2022; Hu et al., 2024c; Chen et al., 2025a; Shen et al., 2025a), advancing latent-space learning (Rombach et al., 2022; Wang et al., 2024d;b; Liu et al., 2024a), refining model architectures (Ho et al., 2022; Peebles & Xie, 2023; Gu et al., 2025; Wang et al., 2024a; Xue et al., 2024; Chen et al., 2025b), and exploring applications in 3D generation (Poole et al., 2022; Wang et al., 2024c; Xu et al., 2024b; Cao et al., 2025a).

## 3. Preliminary

In this section, we introduce the fundamental definitions of our work. In Section 3.1, we introduce all related math notations used in this paper. In Section 3.2, we introduce the key transformer blocks. In Section 3.3, we introduce the components in phase one of the VAR Model. In Section 3.4, we mathematically detail the VAR Transformer blocks.

### 3.1. Notations

We denote the $\ell_p$ norm of a vector $x$ by $\|x\|_p$, i.e., $\|x\|_1 := \sum_{i=1}^{n} |x_i|$, $\|x\|_2 := (\sum_{i=1}^{n} x_i^2)^{1/2}$ and $\|x\|_\infty := \max_{i \in [n]} |x_i|$. For a vector $x \in \mathbb{R}^n$, $\exp(x) \in \mathbb{R}^n$ denotes a vector where $\exp(x)_i$ is $\exp(x_i)$ for all $i \in [n]$. For $n > k$, for any matrix $A \in \mathbb{R}^{n \times k}$, we denote the spectral norm of $A$ by $\|A\|$, i.e., $\|A\| := \sup_{x \in \mathbb{R}^k} \|Ax\|_2 / \|x\|_2$. We define the function norm as $\|f\|_\alpha := (\int \|f(X)\|_\alpha^\alpha dX)^{1/\alpha}$ where $f$ is a function. For a matrix $X \in \mathbb{R}^{n_1 n_2 \times d}$, we use $\mathsf{X} \in \mathbb{R}^{n_1 \times n_2 \times d}$ to denote its tensorization, and we only

assume this for letters $X$ and $Y$.

## 3.2. Transformer Blocks

In this section, we define the components in the VAR Transformer.

We first introduce the Softmax unit.

**Definition 3.1** (Softmax). Let $z \in \mathbb{R}^n$. We define Softmax : $\mathbb{R}^n \to \mathbb{R}^n$ satisfying

$$\mathsf{Softmax}(z) := \frac{\exp(z)}{\langle \exp(z), \mathbf{1}_n \rangle}.$$

Here, we define the attention matrix in the VAR Transformer as follows.

**Definition 3.2** (Attention Matrix). Let $W_Q, W_K \in \mathbb{R}^{d \times d}$ denote the model weights. Let $X \in \mathbb{R}^{n \times d}$ denote the representation of the length-$n$ input. Then, we define the attention matrix $A \in \mathbb{R}^{n \times n}$ by, For $i, j \in [n]$,

$$A_{i,j} := \exp(\underbrace{X_{i,*}}_{1 \times d} \underbrace{W_Q}_{d \times d} \underbrace{W_K^\top}_{d \times d} \underbrace{X_{j,*}^\top}_{d \times 1}).$$

With the attention matrix, we now provide the definition for a single layer of Attention.

**Definition 3.3** (Single Attention Layer). Let $X \in \mathbb{R}^{n \times d}$ denote the representation of the length-$n$ sentence. Let $W_V \in \mathbb{R}^{d \times d}$ denote the model weights. As in the usual attention mechanism, the final goal is to output an $n \times d$ size matrix where $D := \mathrm{diag}(A\mathbf{1}_n) \in \mathbb{R}^{n \times n}$. Then, we define attention layer Attn as

$$\mathsf{Attn}(X) := D^{-1}AXW_V.$$

Here we present the definition of the VAR Attention.

**Definition 3.4** (VAR Attention Layer). Let $r \geq 1$ be a positive integer. Let $h_r, w_r$ be two positive integers. Let $\mathsf{X} \in \mathbb{R}^{h_r \times w_r \times d}$ denote the representation of the input token map. Let $W_V \in \mathbb{R}^{d \times d}$ denote the model weights. As in the usual attention mechanism, the final goal is to output an $n \times d$ size matrix where $D := \mathrm{diag}(A\mathbf{1}_n) \in \mathbb{R}^{h_r w_r \times h_r w_r}$. Then, we define attention layer $\mathsf{Attn}_r : \mathbb{R}^{h_r w_r \times d} \to \mathbb{R}^{h_r w_r \times d}$ as

$$\mathsf{Attn}_r(X) := D^{-1}AXW_V.$$

We introduce the feed-forward layer in the VAR Transformer as follows.

**Definition 3.5** (Single Feed-Forward Layer). If the following conditions hold:

- $X \in \mathbb{R}^{d \times L}$

- $k \in [n]$

- $c$ is the number of neurons

- $W^{(1)} \in \mathbb{R}^{c \times d}, W^{(2)} \in \mathbb{R}^{d \times c}$ are weight matrices

- $b^{(1)} \in \mathbb{R}^c, b^{(2)} \in \mathbb{R}^d$ are bias vectors.

- $\mathsf{FFN} : \mathbb{R}^{d \times L} \to \mathbb{R}^{d \times L}$

Then we define the FFN as follows:

$$\mathsf{FFN}(X)_{*,k} = \underbrace{X_{*,k}}_{d \times 1} + \underbrace{W^{(2)}}_{d \times c} \mathsf{ReLU}(\underbrace{W^{(1)} X_{*,k}}_{c \times 1} + \underbrace{b^{(1)}}_{c \times 1}) + \underbrace{b^{(2)}}_{d \times 1}.$$

## 3.3. VAR **Phase One**

We first present Phase One of VAR model based on (Ke et al., 2025a).

The VAR model uses the VAR Transformer to convert the initialized token map $X_{\mathrm{init}}$ into a series of pyramid-shaped token maps. The VAR Transformer alternates between up sample blocks and attention layers to get the output.

**Up Sample Blocks.** The $k$-th up sample block takes as input the initial token map $X_{\mathrm{init}}$ and the previous pyramid-shaped token maps $X_1, \ldots, X_k$, sets $Y_1 = X_{\mathrm{init}}$ and up samples each $X_i$ into a new token map $Y_{i+1}$, and outputs the new pyramid-shaped token maps $Y_1, \ldots, Y_{k+1}$.

The upsampling on each token map $X_r (r \in [k])$ uses interpolation with a bicubic spline kernel.

**Definition 3.6** (Bicubic Spline Kernel, Definition 3.1 from (Ke et al., 2025a) on Page 5 ). A bicubic spline kernel is a piecewise cubic function $W : \mathbb{R} \to \mathbb{R}$ that satisfies $W(x) \in [0, 1]$ for all $x \in \mathbb{R}$.

We define the up-interpolation layer for one-step geometric series as follows.

**Definition 3.7** (Up-interpolation Layer for One-Step Geometric Series, Definition 3.2 from (Ke et al., 2025a) on Page 5). We let $r \geq 2$ be an integer denoting the interpolation level, and we let $h_{r-1}, w_{r-1}, h_r, w_r \in \mathbb{N}$ satisfy $h_{r-1} < h_r$ and $w_{r-1} < w_r$. We take $d \in \mathbb{N}$ to denote the number of feature channels, and we define the input feature map $\mathsf{X} \in \mathbb{R}^{h_{r-1} \times w_{r-1} \times d}$ and the output feature map $\mathsf{Y} \in \mathbb{R}^{h_r \times w_r \times d}$.

Let $W : \mathbb{R} \to \mathbb{R}$ denote a bicubic spline kernel (see Definition 3.6). The up-interpolation operation is defined by a function

$$\phi_{\mathrm{up},r} : \mathbb{R}^{h_{r-1} \times w_{r-1} \times d} \to \mathbb{R}^{h_r \times w_r \times d}$$

such that $\mathsf{Y} = \phi_{\mathrm{up},r}(\mathsf{X})$. For each spatial index $i \in [h_r]$, $j \in [w_r]$, and channel index $l \in [d]$, the output is given by:

$$\mathsf{Y}_{i,j,l} := \sum_{s=-1}^{2} \sum_{t=-1}^{2} W(s) \cdot \mathsf{X}_{\frac{i \cdot h_{r-1}}{h_r} + s, \frac{j \cdot w_{r-1}}{w_r} + t, \, l} \cdot W(t).$$

After defining the Up-Interpolation Layer for a one-step geometric sequence, we can construct a Pyramid Up-Interpolation Layer, which applies multiple up-interpolation layers to generate token maps at different resolutions. Specifically, we can describe this Pyramid Up-Interpolation Layer through the following definition:

**Definition 3.8** (Pyramid Up-Interpolation Layer $\Phi$, $r = 1$ Case, Definition 3.3 from (Ke et al., 2025a) on Page 5). We let $d \in \mathbb{N}$ be a positive integer denoting the channel dimension, and we take $\mathsf{X}_{\mathrm{init}} \in \mathbb{R}^{1 \times 1 \times d}$ to denote the initial token map.

The pyramid up-interpolation operator at level $r = 1$ is defined as follows:

$$\Phi_{\mathrm{up},1} : \mathbb{R}^{1 \times 1 \times d} \to \mathbb{R}^{1 \times 1 \times d}, \quad \Phi_{\mathrm{up},1}(\mathsf{X}_{\mathrm{init}}) := \mathsf{X}_{\mathrm{init}}.$$

**Definition 3.9** (Pyramid Up-Interpolation Layer $\Phi$, $r \geq 2$ Case). Let $d \in \mathbb{N}$ be the channel dimension, and let $r \geq 2$ be the number of interpolation levels. Let $\mathsf{X}_{\mathrm{init}} \in \mathbb{R}^{1 \times 1 \times d}$ denote the initial token map. Let

$$\phi_{\mathrm{up},i} : \mathbb{R}^{h_{i-1} \times w_{i-1} \times d} \to \mathbb{R}^{h_i \times w_i \times d}$$

denote the up-interpolation operator at level $i$, as defined in Definition 3.7.

We define the pyramid up-interpolation operator

$$\Phi_{\mathrm{up},r} : \mathbb{R}^{h_{[r-1]} \times w_{[r-1]} \times d} \to \mathbb{R}^{h_{[r]} \times w_{[r]} \times d}$$

recursively as follows:

$$\mathsf{Y}_1 = \mathsf{X}_{\mathrm{init}},$$
$$\mathsf{Y}_i = \phi_{\mathrm{up},i-1}(\mathsf{Y}_{i-1}), \quad \text{for all } i \in \{2, \dots, r\}.$$

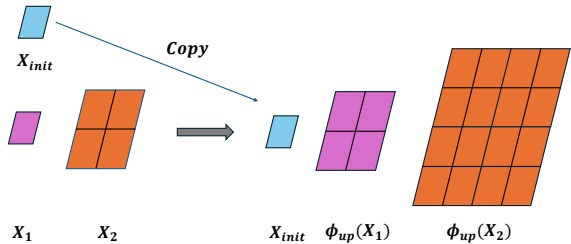

*Figure 1.* One Pyramid Up-Interpolation Layer Instance $\Phi_{\mathrm{up},2}$. The Pyramid Up-Interpolation layer starts from a $1 \times 1 \times d$ token $X_{\mathrm{init}}$ and uses two bicubic up-sampling steps to produce higher-resolution token maps $X_1$ and $X_2$. The concatenated tokens are flattened, and their query–key dot products form the attention matrix $A$ that supplies context to the next transformer block.

*Remark* 3.10. We have a pyramid-shaped token maps of size $h_{[r+1]} \times w_{[r+1]} \times d$. To input this into the VAR Transformer, we merge the first two dimensions, transforming it into an input of size $(\sum_{i=1}^{r+1} h_i w_i) \times d$.

Now, we are ready to introduce the VAR transformer.

**Definition 3.11** (VAR Transformer, Definition 3.1 from (Ke et al., 2025a) on Page 5). If the following conditions hold:

- Assume the VAR transformer has $m$ transformer layers.

- At the $i$-th transformer layer, let $g_i$ denote components excluding the attention layer, such as the LN layer or MLP layer.

- Let $\Phi_{\mathrm{up},r}$ denote the pyramid up-interpolation layer defined in Definition 3.9.

- Let $\mathsf{Attn}_i$ stand for the self-attention layer, which is defined in Definition 3.3.

- Let $\mathsf{X}_{\mathrm{init}} \in \mathbb{R}^{1 \times 1 \times d}$ be an input token map and $X_{\mathrm{init}} \in \mathbb{R}^{1 \times d}$ be its matrix version.

- Let $n = \sum_{i=1}^{m} h_i w_i$.

Then we define a VAR transformer as the following

$$\mathsf{TF}(\mathsf{X}_{\mathrm{init}}) := g_m \circ \mathsf{Attn}_m \circ \Phi_{\mathrm{up},m} \circ \cdots \circ g_2 \circ \mathsf{Attn}_2 \circ \Phi_{\mathrm{up},2}$$
$$\circ\, g_1 \circ \mathsf{Attn}_1 \circ \Phi_{\mathrm{up},1}(\mathsf{X}_{\mathrm{init}}) \in \mathbb{R}^{n \times d},$$

In this expression, $\circ$ stands for functional composition.

### 3.4. VAR **Transformer Blocks**

Recall we have defined $\phi_{\mathrm{up}} : \mathbb{R}^{h \times w \times c} \to \mathbb{R}^{h' \times w' \times c}$ in Definition 3.7. Since there is no non-linear operation in $\phi_{\mathrm{up}}$, $\phi_{\mathrm{up}}$ is equivalent to a matrix multiplication operation, where the dimension of the matrix is $\mathbb{R}^{h'w' \times hw}$. For simplicity, we view $\phi_{\mathrm{up}}$ as a $\mathbb{R}^{h'w' \times hw}$ dimension matrix in the following proofs.

*Remark* 3.12 (Applying $\phi_{\mathrm{up}}$ on $X \in \mathbb{R}^{n \times d}$, Remark 4.8 from (Ke et al., 2025a) on Page 8). The actual input of VAR Transformer Layer are $r$ input token maps, $X_1 \in \mathbb{R}^{h_1 \times w_1 \times d}, \dots, X_r \in \mathbb{R}^{h_r \times w_r \times d}$. We denote them as $X \in \mathbb{R}^{n \times d}$, where $n := \sum_{i=1}^{r} h_i w_i$. We denote $\phi_{\mathrm{up}}(X) \in \mathbb{R}^{n' \times d}$ as applying $\phi_{\mathrm{up}}$ to each $X_i \in \mathbb{R}^{h_i \times w_i \times d}$ for $i \in [r]$, where $n' = \sum_{i=1}^{r} h_i' w_i'$.

Then, we can combine multiple attention layers with other components (up-interpolation layers, multilayer perceptron layers, layer-wise normalization layers) to create a complete VAR Transformer architecture.

**Definition 3.13** (Single VAR Transformer Layer, Definition 4.9 from (Ke et al., 2025a) on Page 9). If the following conditions hold:

- Assume the VAR transformer has $m$ Transformer layers.

- Let FFN denotes a single Feed-forward Layer (see Definition 3.5).

- Let Attn stands for a single self-attention layer (see Definition 3.3).

Then we define a VAR transformer block as the following.

$$\mathsf{TF}_{\mathsf{VAR}}(X) = \mathsf{FFN} \circ \mathsf{Attn} \circ \phi_{\mathrm{up}} \in \mathbb{R}^{n \times d},$$

In this expression, $\circ$ stands for functional composition.

Now, we present the VAR Transformer Network Function Class.

**Definition 3.14** (VAR Transformer Network Function Class). If the following conditions hold:

- Assume the VAR transformer network has $m$ layers.

- for $i \in [m]$, $\mathsf{FFN}_i$ denotes the Feed-forward at $i$-th layer (see Definition 3.5), $\mathsf{Attn}_i$ denotes the Attention at $i$-th layer (see Definition 3.3), and $\phi_i^{\mathrm{up}}$ denotes the Up interpolation at $i$-th layer (see Definition 3.7).

- Let $\mathcal{T}^{a,s,c}$ denote the VAR transformer network function class

- each function $\tau \in \mathcal{T}^{a,s,c}$ consists of VAR transformer blocks $\mathsf{TF}_{\mathsf{VAR}}^m$ with $a$ heads of size $s$ and $c$ MLP hidden neurons

Then we define VAR Transformer Network Function Class as follows:

$$\mathcal{T}^{a,s,c} := \{\tau : \mathbb{R}^{n \times d} \to \mathbb{R}^{n \times d}$$
$$| \tau = \mathsf{TF}_{\mathsf{VAR}}^m \circ \mathsf{TF}_{\mathsf{VAR}}^{m-1} \circ \ldots \circ \mathsf{TF}_{\mathsf{VAR}}^1(X)\}$$

## 4. Main Results

In this section, we introduce our main results. We show that VAR Transformer and FlowAR models are universal approximators.

We first present the result of the universal approximation property of VAR Transformer.

**Theorem 4.1** (Universality of VAR Transformer, Informal Version of Theorem 6.6). *For any word-to-image function* $f_{\mathrm{word2img}}$ *under some mild assumptions, there exists a* VAR *Transformer* $\tau_{\mathsf{VAR}}$ *which can approximate it.*

Next, we extend the result of VAR Transformer to FlowAR models.

**Corollary 4.2** (Universality of FlowAR, Informal Version of Corollary 7.2). *For any word-to-image function* $f_{\mathrm{word2img}}$ *under some mild assumptions, there exists a FlowAR model* $\tau_{\mathsf{VAR}}$ *which can approximate it.*

Theorem 4.1 and Corollary 4.2 jointly reveal that minimal instantiations of both VAR Transformers and FlowAR models already possess universal approximation power. This finding is striking for two reasons. First, it shows that the coarse-to-fine "next-scale prediction" schedule of VAR, when coupled with a single-head self-attention mechanism, suffices to match the expressiveness guarantees previously reserved for much deeper or wider transformer stacks. Second, the same argument extends seamlessly to FlowAR, indicating that invertible flow steps do not diminish expressivity and can even complement autoregressive attention by offering stable training dynamics. Practically, these results suggest that future work can focus on efficiency optimisations—such as weight sharing or low-rank adaptation—without risking a loss of representational capacity.

## 5. Any-Rank Single-Layer Attention is a Contextual Mapping Function

In this section, we show that Attention is a contextual mapping function. In Section 5.1, we give the definition of contextual mapping. In Section 5.2, we introduce any-rank single-layer attention as a contextual mapping function.

### 5.1. Contextual Mapping

**Contextual Mapping.** Let $X, Y \in \mathbb{R}^{n \times d}$ be the input embeddings and output label sequences, respectively. Let $X_i \in \mathbb{R}^d$ be the $i$-th token of each $X$ embedding sequence.

**Definition 5.1** (Vocabulary, Definition 2.4 from (Hu et al., 2024b) on Page 8). We define the vocabulary.

- We define the $i$-th vocabulary set for $i \in [N]$ by $\mathcal{V}^{(i)} = \cup_{k \in [n]} X_k^{(i)} \subset \mathbb{R}^d$.

- We define the whole vocabulary set $\mathcal{V}$ as $\mathcal{V} = \cup_{i \in [N]} \mathcal{V}^{(i)} \subset \mathbb{R}^d$.

Note that while "vocabulary" typically refers to the tokens' codomain, here, it refers to the set of all tokens within a single sequence. To facilitate our analysis, we introduce the idea of input token separation following (Kajitsuka & Sato, 2024; Kim et al., 2022; Yun et al., 2020).

**Definition 5.2** (Tokenwise Separateness, Definition 2.5 from (Hu et al., 2024b) on Page 8). We define the tokenwise separateness as follows.

- Let $X^{(1)}, \ldots, X^{(N)} \in \mathbb{R}^{n \times d}$ be embeddings.

- Let $N$ be the number of sequences in the datasets.

- Let $n$ be the length of a sequence. i.e. $X^{(i)} \in \mathbb{R}^{n \times d}$

First, we state three conditions for $X^{(1)}, \ldots, X^{(N)}$

(i) For any $i \in [N]$ and $k \in [n]$, $\|X_k^{(i)}\|_2 > \gamma_{\min}$ holds.

(ii) For any $i \in [N]$ and $k \in [n]$, $\|X_k^{(i)}\|_2 < \gamma_{\max}$ holds.

(iii) For any $i, j \in [N]$ and $k, l \in [n]$ if $X_k^{(i)} \neq X_l^{(j)}$, then $\|X_k^{(i)} - X_l^{(j)}\|_2 > \delta$ holds.

Second, we define three types of separateness as follows,

- **Part 1.** If all conditions hold, then we call it tokenwise $(\gamma_{\min}, \gamma_{\max}, \delta)$-separated

- **Part 2.** If conditions (ii) and (iii) hold, then we denote this as $(\gamma, \delta)$-separateness.

- **Part 3.** If only condition (iii) holds, then we denote it as $(\delta)$-separateness.

To clarify condition (iii), we consider cases where there are repeated tokens between different input sequences. Next, we define contextual mapping. Contextual mapping describes a function's ability to capture the context of each input sequence as a whole and assign a unique ID to each input sequence.

**Definition 5.3** $((\gamma, \delta)$-Contextual Mapping, Definition 2.6 from (Hu et al., 2024b) on Page 8)**.** A function $q : \mathbb{R}^{n \times d} \to \mathbb{R}^{n \times d}$ is said to be a $(\gamma, \delta)$-contextual mapping for a set of embeddings $X^{(1)}, \ldots, X^{(N)} \in \mathbb{R}^{n \times d}$, if the following conditions hold:

- **Contextual Sensitivity** $\gamma$**.** For any $i \in [N]$ and $k \in [n]$, $\|q(X^{(i)})_k\|_2 < \gamma$ holds.

- **Approximation Error** $\delta$**.** For any $i, j \in [N]$ and $k, l \in [n]$ such that $\mathcal{V}^{(i)} \neq \mathcal{V}^{(j)}$ or $X_k^{(i)} \neq X_l^{(j)}$, $\|q(X^{(i)})_k - q(X^{(j)})_l\|_2 > \delta$ holds.

In addition, Note that $q(X^{(i)})$ for $i \in [N]$ is called a context ID of $X^{(i)}$.

### 5.2. Any-Rank Single-Layer Attention is a Contextual Mapping Function

Now we present the result showing that a softmax-based 1-head, 1-layer attention block with any-rank weight matrices is a contextual mapping.

**Lemma 5.4** (Any-Rank Attention as a $(\gamma, \delta)$-Contextual Mapping, Lemma 2.2 from (Hu et al., 2024b) on Page 9)**.** *If the following conditions hold:*

- *Let $X^{(1)}, \ldots, X^{(N)} \in \mathbb{R}^{n \times d}$ be embeddings that are $(\gamma_{\min}, \gamma_{\max}, \epsilon)$-tokenwise separated, with the vocabulary set $\mathcal{V} = \cup_{i \in [N]} \mathcal{V}^{(i)} \subset \mathbb{R}^d$.*

- *$X_k^{(i)} \neq X_l^{(i)}$ for any $i \in [N]$ and $k, l \in [L]$.*

- *Let $\gamma = \gamma_{\max} + \frac{\epsilon}{4}$*

- *Let $\delta = \exp(-5\epsilon^{-1}|\mathcal{V}|^4 d\kappa\gamma_{\max} \log L)$*

- *Let $\kappa := \gamma_{\max}/\gamma_{\min}$.*

- *Let $W^{(O)} \in \mathbb{R}^{d \times s}$ and $W_V, W_K, W_Q \in \mathbb{R}^{s \times d}$.*

*Then, we can show*

- *1-layer, single-head attention mechanism serves as a $(\gamma, \delta)$-contextual mapping for the embeddings $X^{(1)}, \ldots, X^{(N)}$ with weight matrices $W^{(O)}$ and $W_V, W_K, W_Q$.*

Lemma 5.4 indicates that any-rank self-attention function distinguishes input tokens $X_k^{(i)} = X_l^{(j)}$ such that $\mathcal{V}^{(i)} \neq \mathcal{V}^{(j)}$. In other words, it distinguishes two identical tokens within a different context.

### 5.3. Discussion

Hu et al. (2024c) rigorously studied the statistical and computational frontiers of prompt-tuning Transformers. They prove that even the minimal architecture—a single-layer, single-head Transformer kept entirely frozen—becomes a universal approximator once a learnable soft-prompt is prepended, and they quantify the accompanying memorization cost by showing an exponential lower bound on the prompt length required for arbitrary sequence-to-sequence datasets. Our study builds on this insight by exploring how the unversal approximation abilities of VAR Transformers.

## 6. Universality of VAR Transformer

In this section, we present our proof for the universality of the VAR Transformer. In Section 6.1, we used a universality result from a previous work. In Section 6.2, we analyze how the error behaves when two consecutive layers in our composition are each replaced by their respective approximations. In Section 6.3, we present the scenario when one of the composited layers got replaced by a different function. In Section 6.4, we present the scenario when all of the composited layers got replaced. In Section 6.5, we present our proof for the universality of the VAR Transformer.

### 6.1. Universality of $\mathcal{T}_A^{1,1,4}$ with $O((1/\epsilon)^{dn})$ FFN Layers

We follow the line of the universal approximation techniques (Yun et al., 2020; Kim et al., 2022; Kajitsuka & Sato, 2024; Hu et al., 2024b; Liu et al., 2025; Hu et al., 2025a) and extend to the following result.

**Lemma 6.1** ($\tau \in \mathcal{T}_A^{1,1,4}$ Transformer is Universal Img2Img Approximator, a Variation of Theorem 2.3 in Hu et al. (2024b) on Page 11)**.** *If the following conditions hold:*

- *Let $1 \le p < \infty$ and $\epsilon > 0$.*

- *Let a transformer with one self-attention layer defined as $\tau \in \mathcal{T}_A^{1,1,4}$*

*Then, there exists a transformer $\tau$ with single self-attention layer, such that for any $f_{\mathrm{img2img}} \in \mathcal{F}_C$ there exists $\|\tau(\cdot), f_{\mathrm{img2img}}\|_\alpha \le \epsilon$.*

*Proof.* We provide a sketch of the proof here, which mainly follows from Hu et al. (2024b)

- We start by quantizing the input and output domain of $f_{\mathrm{img2img}} \in \mathcal{F}_{C,r}$ into a quantized function $\overline{f}_{\mathrm{img2img}} : \mathcal{G}_{\delta,h_r w_r} \to \mathcal{G}_{\delta,h_r w_r}$ where $\mathcal{G}_{\delta,h_r w_r} = \{0, \delta, 2\delta, \dots, 1-\delta\}^{h_r w_r \times d}$. Here, $\overline{f}_{\mathrm{img2img}}, \overline{\mathcal{F}}_{C,r}$ denote the quantized function and function class. This is basically performing a piece-wise constant approximation with bounded error $\delta$.

- Next, we construct a surrogate quantized image-to-image function

  $$h_{\mathrm{img2img}} : \mathcal{G}_{\delta,(h_r w_r)} \to \mathcal{G}_{\delta,(h_r w_r)},$$
  $$\text{where } \mathcal{G}_{\delta,(h_r w_r)} = \{0, \delta, 2\delta, \dots, 1-\delta\}^{(h_r w_r) \times d}.$$

  Here $h_{\mathrm{img2img}}$ takes embeddings $X$ as inputs. Crucially, its output approximates any $\overline{f}_{\mathrm{img2img}} \in \overline{\mathcal{F}}_{C,r}$.

- Finally, we show that there exist a transformer $\tau \in \mathcal{T}_A^{1,1,4}$ approximating $h_{\mathrm{img2img}}$ to any precision. The formal construction is in Lemma C.1. By simple reduction from $h_{\mathrm{img2img}}$, $\overline{f}_{\mathrm{img2img}}$ and $f_{\mathrm{img2img}}$, we achieve the universality of prompt tuning on $\mathcal{T}_A^{1,1,4}$ with $O((1/\epsilon)^{d(h_r w_r)})$ FFN layers, where $\epsilon$ is the approximation error.

Thus we complete the proof □

## 6.2. Two Layers Perturbation

In this section, we analyze how the error behaves when two consecutive layers in our composition are each replaced by their respective approximations. Specifically, we consider the composition $f_i \circ g_i$ and replace $g_i$ with an up interpolation function $\Phi_{\mathrm{up},i}$ and $f_i$ with a one-layer transformer $\tau_i$. We show that under appropriate Lipschitz and approximation assumptions, the overall error of the approximated two-layer composition can be controlled in terms of the individual approximation errors.

**Assumption 6.2** (Target Function Class). We assume the following things:

- Let $f_1, \dots, f_r$ be $r$ $K$-Lipschitz functions from $\mathbb{R}^{h_r \times w_r \times d}$ to $\mathbb{R}^{h_r \times w_r \times d}$.

- For each $i \in [r]$, let $g_i$ be a $K$-Lipschitz function from $\mathbb{R}^{h_{i-1} \times w_{i-1} \times d}$ to $\mathbb{R}^{h_i \times w_i \times d}$.

- We assume that for each $i \in [r]$, $g_i$ can be approximated by some up interpolation function $\phi_{\mathrm{up},i}$.

- We assume that the target function $f_{\mathrm{word2img}}$ : $\mathbb{R}^{1 \times 1 \times d} \to \mathbb{R}^{h_r \times w_r \times d}$ satisfies

$$f_{\mathrm{word2img}} := f_r \circ g_r \cdots \circ f_1 \circ g_1.$$

With the Assumption 6.2, we present the two layers of perturbation as follows.

**Lemma 6.3** (Two Layers Perturbation). *Let $\phi_{\mathrm{up},i}$ be the up interpolation function defined in Definition 3.7. Let $f_r$ be $r$ $K$-Lipschitz functions from Assumption 6.2. Let $g_i$ be $r$ $K$-Lipschitz functions from Assumption 6.2. Let $\tau_i$ be the one-layer transformer defined in Eq. 2.4 from (Hu et al., 2024b). If $\|g_i - \Phi_{\mathrm{up},i}\| \le \epsilon_{1,i}$ from Assumption 6.2, $\|f_i - \tau_i\| \le \epsilon_{2,i}$ from Theorem 6.1, and each $f_i$ is $K_{1,i}$-Lipschitz, then we have*

$$\|f_i \circ g_i - \tau_i \circ \Phi_{\mathrm{up},i}\| \le K_{1,i}\epsilon_{1,i} + \epsilon_{2,i}.$$

*Proof.* We can show that

$$
\begin{aligned}
& \|f_i \circ g_i - \tau_i \circ \Phi_{\mathrm{up},i}\| \\
= {} & \|f_i \circ g_i - f_i \circ \Phi_{\mathrm{up},i} + f_i \circ \Phi_{\mathrm{up},i} - \tau_i \circ \Phi_{\mathrm{up},i}\| \\
\le {} & \|f_i \circ g_i - f_i \circ \Phi_{\mathrm{up},i}\| + \|f_i \circ \Phi_{\mathrm{up},i} - \tau_i \circ \Phi_{\mathrm{up},i}\| \\
= {} & \|f_i \circ (g_i - \Phi_{\mathrm{up},i})\| + \|(f_i - \tau_i) \circ \Phi_{\mathrm{up},i}\| \\
\le {} & \|f_i \circ (g_i - \Phi_{\mathrm{up},i})\| + \|f_i - \tau_i\| \\
\le {} & K_{1,i}\epsilon_{1,i} + \epsilon_{2,i}
\end{aligned}
$$

where the first step follows from basic algebra, the second step follows from triangle inequality, the third and fourth steps follow from basic algebra, and the fifth step follows from our conditions. □

## 6.3. Perturbation of Recursively Composting Functions that One Layer is Different

In this section, we consider a scenario where we have a composition of many layers, but only one of the layers is replaced by a different function. This setting helps us see how a single local perturbation can propagate through subsequent layers in a multi-layer composition. The lemma below quantifies this propagation by leveraging Lipschitz continuity.

**Lemma 6.4** (Perturbation of Recursively Composting Functions, One Layer is Different). *if the following conditions hold*

- *Assume $\|u_j(w) - v_j(w)\| \le \epsilon$ for any $w$.*

- *$v_i(x) \le K_2 \cdot \|x\|$*

*Fix $j$, we have*

$$\| \circ_{i=j+1}^{n+1} v_i \circ_{i=1}^{j} u_i - \circ_{i=j}^{n} v_i \circ_{i=0}^{j-1} u_i \| \leq K_2^{n-j} \cdot \epsilon$$

*Proof.* We define $w$ as

$$w = \circ_{i=0}^{j-1} u_i(x)$$

We can show that for any $x$

$$\begin{aligned}
&\| \circ_{i=j+1}^{n+1} v_i \circ_{i=1}^{j} u_i(x) - \circ_{i=j}^{n} v_i \circ_{i=0}^{j-1} u_i(x) \| \\
&= \| \circ_{i=j+1}^{n+1} v_i u_j(w) - \circ_{i=j+1}^{n+1} v_i(v_j(w)) \| \\
&= \| \circ_{i=j+1}^{n+1} v_i(u_j(w) - v_j(w)) \| \\
&\leq K_2^{n-j} \cdot \epsilon
\end{aligned}$$

where the first step follows from basic algebra, the second step follows from linearity, and the third step follows from lemma assumptions. □

### 6.4. Perturbation of Recursively Composting Functions that All Layer are Different

In this section, we extend the analysis to the most general scenario in which all layers in the composition are replaced by different functions. This captures the situation where each layer $u_i$ is approximated by some other function $v_i$. We derive a cumulative bound that sums the individual perturbations introduced at each layer.

**Lemma 6.5** (Perturbation of Recursively Compositing Functions, All Layers are Different)**.** *If the following conditions hold:*

- *Let $\circ_{i=1}^{n} u_i = u_n \circ \cdots \circ u_1$.*

- *Let $\circ_{i=1}^{n} v_i = v_n \circ \cdots \circ v_1$.*

- *Let $u_0(x) = x$ which is identity mapping.*

- *Let $v_{n+1}(x) = x$ which is identity mapping.*

*Then*

$$\begin{aligned}
&\| \circ_{i=1}^{n} u_i - \circ_{i=1}^{n} v_i \| \\
&\leq \sum_{j=1}^{n} \| \circ_{i=j+1}^{n+1} v_i \circ_{i=1}^{j} u_i - \circ_{i=j}^{n} v_i \circ_{i=0}^{j-1} u_i \|
\end{aligned}$$

*Proof.* We can show

$$\begin{aligned}
&\| \circ_{i=1}^{n} u_i - \circ_{i=1}^{n} v_i \| \\
&= \| \sum_{j=1}^{n} (\circ_{i=j+1}^{n+1} v_i \circ_{i=1}^{j} u_i - \circ_{i=j}^{n} v_i \circ_{i=0}^{j-1} u_i) \| \\
&\leq \sum_{j=1}^{n} \| \circ_{i=j+1}^{n+1} v_i \circ_{i=1}^{j} u_i - \circ_{i=j}^{n} v_i \circ_{i=0}^{j-1} u_i \|
\end{aligned}$$

where the first step follows from adding intermediate terms, and the last step follows from the triangle inequality.

Thus, we complete the proof. □

### 6.5. The Universality of VAR Transformer

In this section, with the established error bounds for replacing individual or multiple layers with alternative functions, we now prove the main universality result for the VAR Transformer. In essence, we show that a properly constructed VAR Transformer can approximate the target function $f_{\text{word2img}}$ (from Assumption 6.2) with arbitrarily small errors under suitable Lipschitz and approximation assumptions on each layer.

**Theorem 6.6** (Universality of VAR Transformer, Formal Version of Theorem 4.1)**.** *For $f_{\text{word2img}}$ satisfies Assumption 6.2, there exists a VAR Transformer $\tau_{\text{VAR}}$ such that*

$$\| \tau_{\text{VAR}} - f_{\text{word2img}} \| \leq O(\epsilon)$$

*Proof.* Assume $K_2 > 2$. We can show that

$$\begin{aligned}
\| \tau_{\text{VAR}} - f_{\text{word2img}} \| &= \| \circ_{i=1}^{r} (f_i \circ g_i) - \circ_{i=1}^{r} (\tau_i \circ \Phi_{\text{up},i}) \| \\
&= \sum_{j=1}^{n} K_2^{n-j}(K_{1,i}\epsilon_{1,i} + \epsilon_{2,i}) \\
&= \frac{K_2^n - 1}{K_2 - 1}(K_{1,i}\epsilon_{1,i} + \epsilon_{2,i}) \\
&\leq K_2^n(K_{1,i}\epsilon_{1,i} + \epsilon_{2,i}) \\
&= O(\epsilon).
\end{aligned}$$

where the first step follows from Definition 3.14, the second step follows from Lemma 6.4, the third step follows from basic algebra, and the fourth step follows from the basic inequality, and the last step follows that $\epsilon_{1,i}, \epsilon_{2,i} = O(\epsilon)$ for all $i \in [n]$ □

## 7. Universality of FlowAR and HOFAR

In this section, we show that the universality results established for the VAR Transformer can be extended to FlowAR model. Furthermore, we show that these results also hold for Higher-order FlowAR models (HOFAR) introduced by Liang et al. (2025b). In Section 7.1, we present the universality result of FlowAR. In Section 7.2, we extend the universality result to HOFAR.

### 7.1. Universality of FlowAR

The key observation is that the same local perturbation bounds and Lipschitz assumptions used in the VAR Transformer setting also apply to FlowAR, with only minor changes. Specifically, each FlowAR layer $\Phi_{\text{down},i}$ can be

analyzed in an analogous way to $\Phi_{\text{up},i}$, allowing us to derive a bound on the overall error of the composed FlowAR model.

**Corollary 7.1.** *Let $\phi_{\text{down},i}$ be the down interpolation function of FlowAR (see Definition D.2). Let $f_r$ be $r$ $K$-Lipschitz functions from Assumption 6.2. Let $g_i$ be $r$ $K$-Lipschitz functions from Assumption 6.2. Let $\tau_i \in \mathcal{T}_A^{1,1,4}$ be the one-layer transformer. If $\|g_i - \Phi_{\text{down},i}\| \leq \epsilon_{1,i}$, $\|f_i - \tau_i\| \leq \epsilon_{2,i}$, and each $f_i$ is $K_{1,i}$-Lipschitz, then we have*

$$\|f_i \circ g_i - \tau_i \circ \Phi_{\text{down},i}\| \leq K_{1,i}\epsilon_{1,i} + \epsilon_{2,i}.$$

*Proof.* We can show that

$$\begin{aligned}
&\|f_i \circ g_i - \tau_i \circ \Phi_{\text{down},i}\| \\
=\ &\|f_i \circ g_i - f_i \circ \Phi_{\text{down},i} + f_i \circ \Phi_{\text{down},i} - \tau_i \circ \Phi_{\text{down},i}\| \\
\leq\ &\|f_i \circ g_i - f_i \circ \Phi_{\text{down},i}\| + \|f_i \circ \Phi_{\text{down},i} - \tau_i \circ \Phi_{\text{down},i}\| \\
=\ &\|f_i \circ (g_i - \Phi_{\text{down},i})\| + \|(f_i - \tau_i) \circ \Phi_{\text{down},i}\| \\
\leq\ &\|f_i \circ (g_i - \Phi_{\text{down},i})\| + \|f_i - \tau_i\| \\
\leq\ &K_{1,i}\epsilon_{1,i} + \epsilon_{2,i}
\end{aligned}$$

where the first step follows from basic algebra, the second step follows from triangle inequality, the third and fourth steps follow from basic algebra, and the fifth step follows from our conditions. $\square$

The proof of this corollary mirrors the two-layer perturbation argument from the VAR Transformer, except each "up" interpolation function $\Phi_{\text{up},i}$ is replaced by the corresponding "down" interpolation function $\Phi_{\text{down},i}$. The same Lipschitz and approximation assumptions allow us to bound the difference between $f_i \circ g_i$ and $\tau_i \circ \Phi_{\text{down},i}$.

Next, we can derive the universality of FlowAR models.

**Corollary 7.2** (Universality of FlowAR, Informal Version of Corollary 4.2). *For $f_{\text{word2img}}$ satisfies Assumption 6.2, there exists a FlowAR model $\tau_{\text{FlowAR}}$ such that*

$$\|\tau_{\text{FlowAR}} - f_{\text{word2img}}\| \leq O(\epsilon).$$

*Proof.* Assume $K_2 > 2$. We can show that

$$\begin{aligned}
\|\tau_{\text{FlowAR}} - f_{\text{word2img}}\| &= \|\circ_{i=1}^r (f_i \circ g_i) - \circ_{i=1}^r (\tau_i \circ \Phi_{\text{up},i})\| \\
&= \sum_{j=1}^n K_2^{n-j}(K_{1,i}\epsilon_{1,i} + \epsilon_{2,i}) \\
&= \frac{K_2^n - 1}{K_2 - 1}(K_{1,i}\epsilon_{1,i} + \epsilon_{2,i}) \\
&\leq K_2^n(K_{1,i}\epsilon_{1,i} + \epsilon_{2,i}) \\
&= O(\epsilon).
\end{aligned}$$

where the first step follows from Definition 3.14, the second step follows from Lemma 6.4, the third step follows from

basic algebra, and the fourth step follows from the basic inequality, and the last step follows that $\epsilon_{1,i}, \epsilon_{2,i} = O(\epsilon)$ for all $i \in [n]$ $\square$

The proof of Corollary 7.2 follows the same high-level structure as our universality results for the VAR Transformer (Theorem 6.6. By applying the local perturbation bound layer by layer and then summing the resulting errors, we obtain a global approximation guarantee that is $O(\epsilon)$. Hence, FlowAR, just like the VAR Transformer, can universally approximate the target function $f_{\text{word2img}}$ under the given Lipschitz and approximation assumptions.

### 7.2. Universality of HOFAR

Furthermore, we can extend the result to HOFAR (Liang et al., 2025b). In the original VAR analysis, the attention module learns zeroth-order information (spatial positions). FlowAR upgrades this to first-order information by regressing the velocity field. HOFAR further extends the idea: one attention block predicts the velocity $v^{(1)}$, a second predicts its time derivative $v^{(2)}$, and the two predictions are added—not concatenated—to form the total velocity used for the flow update. Because each block is $K$-Lipschitz, their individual errors add linearly. Hence, if each block is $\epsilon$-accurate, the combined model is $2\epsilon$-accurate, preserving the universal approximation guarantee enjoyed by single-order FlowAR. Formally, we can obtain the following result.

**Corollary 7.3** (Universality of HOFAR). *For $f_{\text{word2img}}$ satisfies Assumption 6.2, there exists a HOFAR model $\tau_{\text{HOFAR}}$ such that*

$$\|\tau_{\text{HOFAR}} - f_{\text{word2img}}\| \leq O(\epsilon).$$

## 8. Conclusion

In this paper, we established that both VAR Transformers and FlowAR architectures serve as universal approximators for Lipschitz sequence-to-sequence functions—even in their most minimal configurations. By dissecting the roles of self-attention, multi-scale up-sampling, and invertible flow transformations, we showed how these components collectively endow the models with sufficient expressive power to capture arbitrary continuous mappings. Our results unify previous theoretical findings on transformer universality with the practical enhancements brought by VAR and flow-based designs, providing a deeper theoretical underpinning for their empirical successes in high-quality image generation and structured prediction tasks. We hope that these findings inspire new explorations into more advanced architectural variants and guide future work on balancing model efficiency, interpretability, and expressive power.

## Acknowledgements

The author would like to thank the anonymous reviewer of ICML 2025 for their highly insightful suggestions.

## Impact Statement

This paper presents work whose goal is to advance the field of Machine Learning. There are many potential societal consequences of our work, none which we feel must be specifically highlighted here.

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

# Appendix

**Roadmap** In Section A, we provide basic algebras that support our proofs. In Section B, we provide other phases of the VAR Model. In Section C, we provide the transformer construction. In Section D, we give the definitions for training the FlowAR Model. In Section E, we give the definitions for the inference of the FlowAR Model. In Section F, we give the definitions for HOFAR. In Section G, we introduce more related work.

## A. Preliminary

In this section, we introduce notations and basic facts that are used in our work. We first list some basic facts of matrix norm properties.

### A.1. Notations

We denote the $\ell_p$ norm of a vector $x$ by $\|x\|_p$, i.e., $\|x\|_1 := \sum_{i=1}^n |x_i|$, $\|x\|_2 := (\sum_{i=1}^n x_i^2)^{1/2}$ and $\|x\|_\infty := \max_{i \in [n]} |x_i|$. For a vector $x \in \mathbb{R}^n$, $\exp(x) \in \mathbb{R}^n$ denotes a vector where $\exp(x)_i$ is $\exp(x_i)$ for all $i \in [n]$. For $n > k$, for any matrix $A \in \mathbb{R}^{n \times k}$, we denote the spectral norm of $A$ by $\|A\|$, i.e., $\|A\| := \sup_{x \in \mathbb{R}^k} \|Ax\|_2 / \|x\|_2$. We define the function norm as $\|f\|_\alpha := (\int \|f(X)\|_\alpha^\alpha dX)^{1/\alpha}$ where $f$ is a function. We use $\sigma_{\min}(A)$ to denote the minimum singular value of $A$. Given two vectors $x, y \in \mathbb{R}^n$, we use $\langle x, y \rangle$ to denote $\sum_{i=1}^n x_i y_i$. Given two vectors $x, y \in \mathbb{R}^n$, we use $x \circ y$ to denote a vector that its $i$-th entry is $x_i y_i$ for all $i \in [n]$. We use $e_i \in \mathbb{R}^n$ to denote a vector where $i$-th entry is 1, and all other entries are 0. Let $x \in \mathbb{R}^n$ be a vector. We define $\mathrm{diag}(x) \in \mathbb{R}^{n \times n}$ as the diagonal matrix whose diagonal entries are given by $\mathrm{diag}(x)_{i,i} = x_i$ for $i = 1, \ldots, n$, and all off-diagonal entries are zero. For a symmetric matrix $A \in \mathbb{R}^{n \times n}$, we say $A \succ 0$ (positive definite (PD)), if for all $x \in \mathbb{R}^n \setminus \{\mathbf{0}_n\}$, we have $x^\top A x > 0$. For a symmetric matrix $A \in \mathbb{R}^{n \times n}$, we say $A \succeq 0$ (positive semidefinite (PSD)), if for all $x \in \mathbb{R}^n$, we have $x^\top A x \geq 0$. The Taylor Series for $\exp(x)$ is $\exp(x) = \sum_{i=0}^\infty \frac{x^i}{i!}$. For a matrix $X \in \mathbb{R}^{n_1 n_2 \times d}$, we use $\mathsf{X} \in \mathbb{R}^{n_1 \times n_2 \times d}$ to denote its tensorization, and we only assume this for letters $X$ and $Y$.

### A.2. Basic Algebra

In this section, we introduce the basic algebras used in our work.

*Fact* A.1. Let $A$ denote the matrix. For each $i$, we use $A_{i,*}$ to denote the $i$-th row of $A$. For $j$, we use $A_{*,j}$ to denote the $j$-th column of $A$. We can show that

- $\|A\| \leq \|A\|_F$

- $\|A\| \geq \|A_{i,*}\|_2$

- $\|A\| \geq \|A_{*,j}\|_2$

Then, we introduce some useful inner product properties.

*Fact* A.2. For vectors $u, v, w \in \mathbb{R}^n$. We have

- $\langle u, v \rangle = \langle u \circ v, \mathbf{1}_n \rangle$

- $\langle u \circ v, w \rangle = \langle u \circ v \circ w, \mathbf{1}_n \rangle$

- $\langle u, v \rangle = \langle v, u \rangle$

- $\langle u, v \rangle = u^\top v = v^\top u$

Now, we show more vector properties related to the hadamard products, inner products, and diagnoal matrices.

*Fact* A.3. For any vectors $u, v, w \in \mathbb{R}^n$, we have

- $u \circ v = v \circ u = \mathrm{diag}(u) \cdot v = \mathrm{diag}(v) \cdot u$

- $u^\top (v \circ w) = u^\top \mathrm{diag}(v) w$

- $u^\top(v \circ w) = v^\top(u \circ w) = w^\top(u \circ v)$

- $u^\top \operatorname{diag}(v)w = v^\top \operatorname{diag}(u)w = u^\top \operatorname{diag}(w)v$

- $\operatorname{diag}(u) \cdot \operatorname{diag}(v) \cdot \mathbf{1}_n = \operatorname{diag}(u)v$

- $\operatorname{diag}(u \circ v) = \operatorname{diag}(u)\operatorname{diag}(v)$

- $\operatorname{diag}(u) + \operatorname{diag}(v) = \operatorname{diag}(u + v)$

# B. VAR Transformer Blocks

### B.1. Phase 2: Feature Map Reconstruction

In this section, we introduce the Phase Two of the VAR model.

**Definition B.1** (Convolution Layer, Definition 3.9 from (Ke et al., 2025a) on Page 9)**.** The Convolution Layer is defined as follows:

- Let $h \in \mathbb{N}$ denote the height of the input and output feature map.

- Let $w \in \mathbb{N}$ denote the width of the input and output feature map.

- Let $c_{\text{in}} \in \mathbb{N}$ denote the number of channels of the input feature map.

- Let $c_{\text{out}} \in \mathbb{N}$ denote the number of channels of the output feature map.

- Let $X \in \mathbb{R}^{h \times w \times c_{\text{in}}}$ denote the input feature map.

- For $l \in [c_{\text{out}}]$, we use $K^l \in \mathbb{R}^{3 \times 3 \times c_{\text{in}}}$ to denote the $l$-th convolution kernel.

- Let $p = 1$ denote the padding of the convolution layer.

- Let $s = 1$ denote the stride of the convolution kernel.

- Let $Y \in \mathbb{R}^{h \times w \times c_{\text{out}}}$ denote the output feature map.

We use $\phi_{\text{conv}} : \mathbb{R}^{h \times w \times c_{\text{in}}} \to \mathbb{R}^{h \times w \times c_{\text{out}}}$ to denote the convolution operation then we have $Y = \phi_{\text{conv}}(X)$. Specifically, for $i \in [h], j \in [w], l \in [c_{\text{out}}]$, we have

$$Y_{i,j,l} := \sum_{m=1}^{3} \sum_{n=1}^{3} \sum_{c=1}^{c_{\text{in}}} X_{i+m-1,j+n-1,c} \cdot K^l_{m,n,c} + b$$

*Remark* B.2. Assumptions of kernel size, padding of the convolution layer, and stride of the convolution kernel are based on the specific implementation of (Tian et al., 2024).

### B.2. Phase 3: VQ-VAE Decoder process

In this section, we introduce Phase Three of the VAR model.

VAR will use the VQ-VAE Decoder Module to reconstruct the feature map generated in Section B.1 into a new image. The Decoder of VQ-VAE has the following main modules (Ke et al., 2025a): (1) Resnet Blocks; (2) Attention Blocks; (3) Up Sample Blocks. We recommend readers to (Ke et al., 2025a) for more details.

# C. Construction of Transformers

In this Section, we show that how to construct $\tau \in \mathcal{T}_A^{1,1,4}$ such that it can approximate any img-to-img function to any precision.

**Lemma C.1** (Transformer Construction, a Variation of Lemma F.2 from (Hu et al., 2024b) on Page 45)**.** *If the following conditions hold:*

- *for any given quantized sequence-to-sequence function*

$$h_{\text{img2img}} : \mathcal{G}_{\delta,(h_r w_r)} \rightarrow \mathcal{G}_{\delta,(h_r w_r)}$$

- *with $\mathcal{G}_{\delta,(h_r w_r)} = \{0, \delta, 2\delta, \ldots, 1 - \delta\}^{(h_r w_r) \times d}$,*

- *Let a transformer be $\tau \in \mathcal{T}_B^{1,1,c}$,*

- *Let positional embedding be $E \in \mathbb{R}^{(h_r w_r) \times d}$*

*Then, there exists*

- *a transformer $\tau$ with positional embedding $E$*

- *such that $d_\alpha(\tau, h(\cdot)) \leq \epsilon/2$.*

*Proof.* First, we apply the positional encoding $E \in \mathbb{R}^{(h_r w_r) \times d}$ on the input sequence $X \in \mathbb{R}^{(h_r w_r + n) \times d}$, so that each token of has a different domain. The positional encoding $E$ is given as

$$E = \begin{bmatrix} 0 & 1 & 2 & \ldots & h_r w_r - 1 \\ 0 & 1 & 2 & \ldots & h_r w_r - 1 \\ \vdots & \vdots & \vdots & \ddots & \vdots \\ 0 & 1 & 2 & \ldots & h_r w_r - 1 \end{bmatrix}.$$

We next use the first feed-forward layer $\mathsf{FFN}_1$ to implement a quantization map to quantize the input $X + E$ into its discrete version $M \in \overline{\mathcal{G}}_\delta$ . Here, we define the grid

$$\overline{\mathcal{G}}_\delta = [\delta : \delta : 1]^d \times [1 + \delta : \delta : 2]^d \times \cdots \times [h_r w_r - 1 + \delta : \delta : h_r w_r]^d,$$

where $[a : \epsilon : b] := \{a, a + \epsilon, a + 2\epsilon, \ldots, b - \epsilon, b\}$. Note that the first column of $X + E$ is in $[0, 1]^d$, the second is in $[1, 2]^d$, and so on. Here, we write the quantization mapping as

$$[0, 1]^d \times \cdots \times [h_r w_r - 1, h_r w_r]^d \rightarrow [\delta : \delta : 1 - \delta]^d \times \cdots \times [h_r w_r - 1 : \delta : h_r w_r]^d,$$

where $[a : \epsilon : b] := \{a, a + \epsilon, a + 2\epsilon, \ldots, b - \epsilon, b\}$. Following (Kajitsuka & Sato, 2024), this quantization task is done by constructing the feed-forward layer as a $\theta$-approximated step function. Consider a real value piece-wise constant function $f^{(\text{Step})} : \mathbb{R} \rightarrow \mathbb{R}$, for any small $\theta > 0, x \in \mathbb{R}$, we have the $\theta$-approximation as

$$
\begin{aligned}
f^{(\text{Step})}(x) &= \lim_{\delta \to 0} \sum_{t=0}^{(h_r w_r)(1/\delta - 1)} (\mathsf{ReLU}(x/\theta - t\delta/\theta) - \mathsf{ReLU}(x/\theta - 1 - t\delta/\theta))\delta \\
&= \lim_{\delta \to 0} \begin{cases} 0, & \text{if } x < 0; \\ \delta, & \text{if } 0 \leq x < \delta; \\ \vdots & \vdots \\ h_r w_r, & \text{if } h_r w_r - \delta \leq x; \end{cases}
\end{aligned}
\tag{1}
$$

which is a series of small step functions, each beginning their rise at $t\delta$ and ending at $\theta + t\delta$. Here, we show the first two terms $t = 0, 1$ for clarity. When $t = 0$, we have

$$(\mathsf{ReLU}(x/\theta) - \mathsf{ReLU}(x/\theta - 1))\delta = \begin{cases} 0, & \text{if } x < 0; \\ x\delta/\theta, & \text{if } 0 \leq x < \theta; \\ \delta, & \text{if } \theta \leq x. \end{cases}$$

When $t = 1$, we have

$$(\mathsf{ReLU}(x/\theta - \delta/\theta) - \mathsf{ReLU}(x/\theta - 1 - \delta/\theta))\delta = \begin{cases} 0, & \text{if } x < \delta; \\ x\delta/\theta, & \text{if } \delta \le x < \theta + \delta; \\ \delta, & \text{if } \theta + \delta \le x. \end{cases}$$

With Eq. (1), it is straightforward that we extend it to $\mathbb{R}^{h_r w_r \times d}$. As a result, we have the first feed-forward layer $\mathsf{FFN}_1$ as

$$\mathsf{FFN}_1(X)_{i,j} = \sum_{t=0}^{(L_p+n)(1/\delta-1)} (\mathsf{ReLU}(X_{i,j}/\theta - t\delta/\theta) - \mathsf{ReLU}(X_{i,j}/\theta - 1 - t\delta/\theta))\delta$$

where $i \in [d], j \in [L_p + n], 0 < \delta < 1$ and $\theta > 0$.

Taking the limit as $\delta \to 0$, we have

$$\lim_{\delta \to 0} \sum_{t=0}^{(L_p+n)(1/\delta-1)} (\mathsf{ReLU}(X_{i,j}/\theta - t\delta/\theta) - \mathsf{ReLU}(X_{i,j}/\theta - 1 - t\delta/\theta))\delta = f^{(\mathrm{Step})}(X_{i,j}). \tag{2}$$

With Eq. (2), we are able to quantize each sequence $X + E$ to a quantized version $M \in \overline{\mathcal{G}}_\delta$ .

Next, in order to utilize Lemma 5.4, we observe that the quantized input $M$ from the previous step has no duplicate tokens, since each column has a unique domain. Also, we see that $M$ is token-wise $(\sqrt{d}, \sqrt{d}(L_0 - \delta), \sqrt{d}\delta)$-separated where $L_0 = h_r w_r$. This is easily observed as we have, for any $k, l \in [h_r w_r]$,

$$\|M_k\|_2 > \sqrt{d},$$
$$\|M_k\|_2 < \sqrt{d}(h_r w_r - \delta),$$
$$\|M_k - L_l\|_2 > \sqrt{d}\delta.$$

As a result, with Lemma 5.4, the single self-attention layer implements a contextual mapping $q : \mathbb{R}^{(h_r w_r) \times d} \to \mathbb{R}^{(h_r w_r) \times d}$, we arrive at a $(\Gamma, \Delta)$-contextual mapping where

$$\Gamma = \sqrt{d}(L_0 - \delta) + \frac{\sqrt{d}\delta}{4} = \sqrt{d}(L_0 - 0.75\delta),$$
$$\Delta = \exp(-5|\mathcal{V}|^4 d \ln(n) L_0^2 / \delta).$$

Now we have successfully mapped each input sequence $[P, X] + E$ to a unique context ID $q(M) \in \mathbb{R}^{(h_r w_r) \times d}$. We next associate each unique embeddings to a corresponding expected output of $h_{\mathrm{img2img}}(\cdot)$.

We associate each unique contextual embeddings to the corresponding output of $h(\cdot)$ using the second feed-forward layer $\mathsf{FFN}_2$. As in Section A.5 of (Kajitsuka & Sato, 2024), this is achieved by constructing a bump function $f_{\mathrm{bump}} : \mathbb{R}^{(h_r w_r) \times d} \to \mathbb{R}^{(h_r w_r) \times d}$ for each possible output from the last step $q(M^{(i)}), i \in [(1/\delta)^{d(h_r w_r)}]$. Each bump function $f_{\mathrm{bump}}$ is realized by $3d(h_r w_r)$ MLP neurons. Therefore, we need $3d(h_r w_r)(1/\delta)^{d(h_r w_r)}$ MLP neurons to construct the feed-forward layer $\mathsf{FFN}_2$, so that each contextual embedding is mapped to the expected output of $h_{\mathrm{img2img}}(\cdot)$. A bump function $f_{\mathrm{bump}}$ for a quantized sequence $A \in \overline{\mathcal{G}}_\delta$ is written as:

$$f_{\mathrm{bump}}(Q) = \frac{h(A)}{d(h_r w_r)} \sum_{i=1}^{d} \sum_{j=1}^{h_r w_r} \left(\mathsf{ReLU}(K(Q_{i,j} - A_{i,j}) - 1) - \mathsf{ReLU}(K(Q_{i,j} - A_{i,j})) + \mathsf{ReLU}(K(Q_{i,j} - A_{i,j}) + 1)\right),$$

where $Q \in \mathbb{R}^{d \times (h_r w_r)}$ is some context ID scalar $K > 0$. Furthermore, we have the relation of quantization granularity $\delta$ and function approximation error $\epsilon$ as $C\delta(dh_r w_r)^{\frac{1}{\alpha}} \le \epsilon/2$. We express the number of neurons in terms of $\epsilon$ as $O(d(h_r w_r)(C(dh_r w_r)^{\frac{1}{\alpha}}/\epsilon)^{d(h_r w_r)}) = O(\epsilon^{-d(h_r w_r)})$, where $C$ is the Lipschitz constant and $\alpha$ is from the $\ell_\alpha$-norm we use for measuring the approximation error.

As a result, by choosing the appropriate step function approximation $\theta$, we arrive at

$$d_p(h_{\mathrm{img2img}}(\cdot), \tau) \le \epsilon/2.$$

This completes the proof. □

# D. Training of FlowAR

In this section, we introduce the training of FlowAR along with its definitions based on (Ren et al., 2024).

In Section D.1, we introduce notation. Section D.2 covers sampling operations, Section D.3 expands them to linear ones. SectionD.4 details the VAE tokenizer, while Section D.5 specifies the transformer backbone. Section D.6 covers flow construction.

## D.1. Notations

We first introduce some notations used in FlowAR.

For a matrix $X \in \mathbb{R}^{n_1 n_2 \times d}$, we use $\mathsf{X} \in \mathbb{R}^{n_1 \times n_2 \times d}$ to denote its tensorization, and we only assume this for letters $X, Y, Z, F, V$.

## D.2. Sample Function

In this section, we introduce the sample functions used in FlowAR.

We first introduce the up sample function.

**Definition D.1** (Up Sample Function). If the following conditions hold:

- Let $h, w \in \mathbb{N}$ denote the height and weight of latent $\mathsf{X} \in \mathbb{R}^{h \times w \times c}$.
- Let $r > 0$ denote a positive integer.

Then we define $\mathrm{Up}(\mathsf{X}, r) \in \mathbb{R}^{rh \times rw \times c}$ as the upsampling of latent $\mathsf{X}$ by a factor $r$.

Then, we introduce the down sample function.

**Definition D.2** (Down Sample Function). If the following conditions hold:

- Let $h, w \in \mathbb{N}$ denote the height and weight of latent $\mathsf{X} \in \mathbb{R}^{h \times w \times c}$.
- Let $r > 0$ denote a positive integer.

Then we define $\mathrm{Down}(\mathsf{X}, r) \in \mathbb{R}^{\frac{h}{r} \times \frac{w}{r} \times c}$ as the downsampling of latent $\mathsf{X}$ by a factor $r$.

## D.3. Linear Sample Function

In this section, we present the linear sample function in FlowAR.

We first present the linear up sample function.

**Definition D.3** (Linear Up Sample Function). If the following conditions hold:

- Let $h, w \in \mathbb{N}$ denote the height and weight of latent $\mathsf{X} \in \mathbb{R}^{h \times w \times c}$.
- Let $r > 0$ denote a positive integer.
- Let $\Phi_{\mathrm{up}} \in \mathbb{R}^{hw \times (rh \cdot rw)}$ be a matrix.

Then we define the linear up sample function $\phi_{\mathrm{up}}(\cdot, \cdot)$ as it computes $\mathsf{Y} := \phi_{\mathrm{up}}(\mathsf{X}, r) \in \mathbb{R}^{rh \times rw \times c}$ such that the matrix version of $\mathsf{X}$ and $\mathsf{Y}$ satisfies

$$Y = \Phi_{\mathrm{up}} X \in \mathbb{R}^{(rh \cdot rw) \times c}.$$

Then, we define linear down sample function as follows.

**Definition D.4** (Linear Down Sample Function). If the following conditions hold:

- Let $h, w \in \mathbb{N}$ denote the height and weight of latent $\mathsf{X} \in \mathbb{R}^{h \times w \times c}$.

- Let $r > 0$ denote a positive integer.

- Let $\Phi_{\text{down}} \in \mathbb{R}^{((h/r) \cdot (w/r)) \times hw}$ be a matrix.

Then we define the linear down sample function $\phi_{\text{down}}(\mathsf{X}, r)$ as it computes $\mathsf{Y} := \phi_{\text{down}}(\mathsf{X}, r) \in \mathbb{R}^{(h/r) \times (w/r) \times c}$ such that the matrix version of $\mathsf{X}$ and $\mathsf{Y}$ satisfies

$$Y = \Phi_{\text{down}} X \in \mathbb{R}^{((h/r) \cdot (w/r)) \times c}.$$

## D.4. VAE Tokenizer

In this section, we show the VAE Tokenizer.

**Definition D.5** (VAE Tokenizer). If the following conditions hold:

- Let $\mathsf{X} \in \mathbb{R}^{h \times w \times c}$ denote a continuous latent representation generated by VAE.

- Let $K$ denote the total number of scales in FlowAR.

- Let $a$ be a positive integer.

- For $i \in [K]$, let $r_i := a^{K-i}$.

- For each $i \in [K]$, let $\phi_{\text{down},i}(\cdot, r_i) : \mathbb{R}^{h \times w \times c} \to \mathbb{R}^{(h/r_i) \times (w/r_i) \times c}$ denote the linear down sample function defined in Definition D.4.

For $i \in [K]$, we define the $i$-th token map generated by VAE Tokenizer be

$$\mathsf{Y}^i := \phi_{\text{down},i}(\mathsf{X}, r_i) \in \mathbb{R}^{(h/r_i) \times (w/r_i) \times c},$$

We define the output of VAE Tokenizer as follows:

$$\mathsf{Tokenizer}(\mathsf{X}) := \{\mathsf{Y}^1, \mathsf{Y}^2, \ldots, \mathsf{Y}^K\}.$$

*Remark* D.6. In (Ren et al., 2024), they choose $a = 2$ and hence for $i \in [n]$, $r_i := 2^{K-i}$.

## D.5. Autoregressive Transformer

Firstly, we give the definition of a single attention layer.

**Definition D.7** (Single Attention Layer). If the following conditions hold:

- Let $h, w \in \mathbb{N}$ denote the height and weight of latent $\mathsf{X} \in \mathbb{R}^{h \times w \times c}$.

- Let $W_Q, W_K, W_V \in \mathbb{R}^{c \times c}$ denote the weight matrix for query, key, and value, respectively.

Then we define the attention layer $\mathsf{Attn}(\cdot)$ as it computes $\mathsf{Y} = \mathsf{Attn}(\mathsf{X}) \in \mathbb{R}^{h \times w \times c}$. For the matrix version, we first need to compute the attention matrix $A \in \mathbb{R}^{hw \times hw}$:

$$A_{i,j} := \exp(X_{i,*} W_Q W_K^\top X_{j,*}^\top), \ \text{ for } i, j \in [hw].$$

Then, we compute the output:

$$Y := D^{-1} A X W_V \in \mathbb{R}^{hw \times c}.$$

where $D := \text{diag}(A \mathbf{1}_n) \in \mathbb{R}^{hw \times hw}$.

To move on, we present the definition of multilayer perceptron.

**Definition D.8** (MLP layer). If the following conditions hold:

- Let $h, w \in \mathbb{N}$ denote the height and weight of latent $\mathsf{X} \in \mathbb{R}^{h \times w \times c}$.

- Let $c$ denote the input dimension of latent $\mathsf{X} \in \mathbb{R}^{h \times w \times c}$.

- Let $d$ denote the dimension of the target output.

- Let $W \in \mathbb{R}^{c \times d}$ denote a weight matrix.

- Let $b \in \mathbb{R}^{1 \times d}$ denote a bias vector.

Then we define mlp layer as it computes $\mathsf{Y} := \mathsf{MLP}(X, c, d) \in \mathbb{R}^{h \times w \times d}$ such that the matrix version of $\mathsf{X}$ and $\mathsf{Y}$ astisfies, for each $j \in [hw]$,

$$Y_{j,:} = \underbrace{X_{j,:}}_{1 \times c} \cdot \underbrace{W}_{c \times d} + \underbrace{b}_{1 \times d}$$

We present the definition of layer-wise norm layer.

**Definition D.9** (Layer-wise norm layer). Given a latent $\mathsf{X} \in \mathbb{R}^{h \times w \times c}$. We define the layer-wise as it computes $\mathsf{Y} := \mathsf{LN}(X) \in \mathbb{R}^{h \times w \times c}$ such that the matrix version of $\mathsf{X}$ and $\mathsf{Y}$ satisfies, for each $j \in [hw]$,

$$Y_{j,:} = \frac{X_{j,:} - \mu_j}{\sqrt{\sigma_j^2}}$$

where $\mu_j := \sum_{k=1}^c X_{j,k}/c$ and $\sigma_j^2 = \sum_{k=1}^c (X_{j,k} - \mu_j)^2/c$.

**Definition D.10** (Autoregressive Transformer). If the following conditions hold:

- Let $\mathsf{X} \in \mathbb{R}^{h \times w \times c}$ denote a continuous latent representation generated by VAE.

- Let $K$ denote the total number of scales in FlowAR.

- For $i \in [K]$, let $\mathsf{Y}_i \in \mathbb{R}^{(h/r_i) \times (w/r_i) \times c}$ be the $i$-th token map genereated by VAE Tokenizer defined in Definition D.5.

- Let $a$ be a positive integer.

- For $i \in [K]$, let $r_i := a^{K-i}$.

- For $i \in [K-1]$, let $\phi_{\mathrm{up},i}(\cdot, a) : \mathbb{R}^{(h/r_i) \times (w/r_i) \times c} \to \mathbb{R}^{(h/r_{i+1}) \times (w/r_{i+1}) \times c}$ be the linear up sample function defined in Definition D.3.

- For $i \in [K]$, let $\mathsf{Attn}_i(\cdot) : \mathbb{R}^{(\sum_{j=1}^i h/r_j)(\sum_{j=1}^i w/r_j) \times c} \to \mathbb{R}^{(\sum_{j=1}^i h/r_j)(\sum_{j=1}^i w/r_j) \times c}$ be the $i$-th attention layer defined in Definition D.7.

- For $i \in [K]$, let $\mathsf{FFN}_i(\cdot) : \mathbb{R}^{(\sum_{j=1}^i h/r_j)(\sum_{j=1}^i w/r_j) \times c} \to \mathbb{R}^{(\sum_{j=1}^i h/r_j)(\sum_{j=1}^i w/r_j) \times c}$ be the $i$-th feed forward network defined in Definition 3.5.

- Let $\mathsf{Z}_{\mathrm{init}} \in \mathbb{R}^{(h/r_1) \times (w/r_1) \times c}$ be the initial input denoting the class condition.

- Let $\mathsf{Z}^1 := \mathsf{Z}_{\mathrm{init}} \in \mathbb{R}^{(h/r_1) \times (w/r_1) \times c}$.

- For $i \in [K] \setminus \{1\}$, Let $\mathsf{Z}^i$ be the reshape of the input sequence $\mathsf{Z}_{\mathrm{init}}, \phi_{\mathrm{up},1}(\mathsf{Y}^1, a), \ldots, \phi_{\mathrm{up},i}(\mathsf{Y}^{i-1}, a)$ into the tensor of size $(\sum_{j=1}^i h/r_j) \times (\sum_{j=1}^i w/r_j) \times c$.

For $i \in [K]$, we define the Autoregressive transformer $\mathsf{TF}_i$ as

$$\mathsf{TF}_i(\mathsf{Z}^i) = \mathsf{FFN}_i \circ \mathsf{Attn}_i(\mathsf{Z}^i) \in \mathbb{R}^{(\sum_{j=1}^i h/r_j)(\sum_{j=1}^i w/r_j) \times c}.$$

We denote $\widehat{\mathsf{Y}}^i$ as the $i$-th block of size $(h/r_i) \times (w/r_i) \times c$ of the tensorization of $\mathsf{TF}_i(\mathsf{Z}^i)$.

### D.6. Flow Matching

In this section, we introduce the flow matching definition.

**Definition D.11** (Flow). If the following conditions hold:

- Let $X \in \mathbb{R}^{h \times w \times c}$ denote a continuous latent representation generated by VAE.

- Let $K$ denote the total number of scales in FlowAR.

- For $i \in [K]$, let $Y_i \in \mathbb{R}^{(h/r_i) \times (w/r_i) \times c}$ be the $i$-th token map genereated by VAE Tokenizer defined in Definition D.5.

- For $i \in [K]$, let $F_0^i \in \mathbb{R}^{(h/r_i) \times (w/r_i) \times c}$ be a matrix where each entry is sampled from the standard Gaussian $\mathcal{N}(0, 1)$.

We defined the interpolated input as follows:

$$F_t^i := tY^i + (1 - t)F_0^i.$$

The velocity flow is defined as

$$V_t^i := \frac{dF_t^i}{dt} = Y^i - F_0^i.$$

Here, we define the architecture of the flow matching model defined in (Ren et al., 2024).

**Definition D.12** (Flow Matching Architecture). If the following conditions hold:

- Let $X \in \mathbb{R}^{h \times w \times c}$ denote a continuous latent representation generated by VAE.

- Let $K$ denote the total number of scales in FlowAR.

- For $i \in [K]$, let $Y_i \in \mathbb{R}^{(h/r_i) \times (w/r_i) \times c}$ be the $i$-th token map genereated by VAE Tokenizer defined in Definition D.5.

- Let $i \in [K]$.

- Let $\widehat{Y}_i \in \mathbb{R}^{(h/r_i) \times (w/r_i) \times c}$ be the $i$-th block of the output of Autoregressive Transformer defined in Definition D.10.

- Let $F_t^i$ be the interpolated input defined in Definition D.11.

- Let $\mathsf{Attn}_i(\cdot) : \mathbb{R}^{(h/r_i) \times (w/r_i) \times c} \to \mathbb{R}^{(h/r_i) \times (w/r_i) \times c}$ be the $i$-th attention layer defined in Definition D.7.

- Let $\mathsf{MLP}_i(\cdot, c, d) : \mathbb{R}^{(h/r_i) \times (w/r_i) \times c} \to \mathbb{R}^{(h/r_i) \times (w/r_i) \times d}$ be the $i$-th attention layer defined in Definition D.8.

- Let $\mathsf{LN}_i(\cdot) : \mathbb{R}^{(h/r_i) \times (w/r_i) \times c} \to \mathbb{R}^{(h/r_i) \times (w/r_i) \times c}$ be the $i$-th layer-wise norm layer defined in Definition D.9.

- Let $t_i \in [0, 1]$ denote a time step.

Then we define the $i$-th flow matching model as $\mathsf{NN}_i(F_t^i, \widehat{Y}_i, t_i) : \mathbb{R}^{(h/r_i) \times (w/r_i) \times c} \times \mathbb{R}^{(h/r_i) \times (w/r_i) \times c} \times \mathbb{R} \to \mathbb{R}^{(h/r_i) \times (w/r_i) \times c}$. The tensor input needs to go through the following computational steps:

- **Step 1:** Compute intermediate variables $\alpha_1, \alpha_2, \beta_1, \beta_2, \gamma_1, \gamma_2$. Specifically, we have

$$\alpha_1, \alpha_2, \beta_1, \beta_2, \gamma_1, \gamma_2 := \mathsf{MLP}_i(\widehat{Y}_i + t_i \cdot \mathbf{1}_{(h/r_i) \times (w/r_i) \times c}, c, 6c)$$

- **Step 2:** Compute intermediate variable $\widehat{F}_t^{i'}$. Specifically, we have

$$\widehat{F}_t^{i'} := \mathsf{Attn}_i(\gamma_1 \circ \mathsf{LN}(F_t^i) + \beta_1) \circ \alpha_1$$

where $\circ$ denotes the element-wise product for tensors.

- **Step 3:** Compute final output $\mathsf{F}_t^{i''}$. Specifically, we have

$$\mathsf{F}_t^{i''} = \mathsf{MLP}_i(\gamma_2 \circ \mathsf{LN}(\widehat{F}_t^{i'}) + \beta_2, c, c) \circ \alpha_2$$

where $\circ$ denotes the element-wise product for tensors.

Then, we present our training objective.

**Definition D.13** (Loss of FlowAR). If the following conditions hold:

- Let $\mathsf{X} \in \mathbb{R}^{h \times w \times c}$ denote a continuous latent representation generated by VAE.

- Let $K$ denote the total number of scales in FlowAR.

- For $i \in [K]$, let $\mathsf{Y}_i \in \mathbb{R}^{(h/r_i) \times (w/r_i) \times c}$ be the $i$-th token map genereated by VAE Tokenizer defined in Definition D.5.

- For $i \in [K]$, let $\widehat{\mathsf{Y}}_i \in \mathbb{R}^{(h/r_i) \times (w/r_i) \times c}$ be the $i$-th block of the output of Autoregressive Transformer defined in Definition D.10.

- For $i \in [K]$, let $\mathsf{F}_t^i$ be the interpolated input defined in Definition D.11.

- For $i \in [K]$, let $\mathsf{V}_t^i$ be the velocity flow defined in Definition D.11.

- For $i \in [K]$, let $\mathsf{NN}_i(\cdot, \cdot, \cdot) : \mathbb{R}^{(h/r_i) \times (w/r_i) \times c} \times \mathbb{R}^{(h/r_i) \times (w/r_i) \times c} \times \mathbb{R} \to \mathbb{R}^{(h/r_i) \times (w/r_i) \times c}$ denote the $i$-th flow matching network defined in Definition D.12.

The loss function of FlowAR is

$$L(\theta) = \sum_{i=1}^{n} \mathbb{E}_{t \sim \mathsf{Unif}[0,1]} \|\mathsf{NN}_i(\mathsf{F}_t^i, \widehat{\mathsf{Y}}_t^i, t_i) - \mathsf{V}_t^i\|^2.$$

# E. Inference of FlowAR

We define the architecture of FlowAR during the inference process as follows.

**Definition E.1** (FlowAR Architecture in the Inference Pipeline). If the following conditions hold:

- Let $K$ denote the total number of scales in FlowAR.

- Let $a$ be a positive integer.

- For $i \in [K]$, let $r_i := a^{K-i}$.

- For $i \in [K-1]$, let $\phi_{\mathrm{up},i}(\cdot, a) : \mathbb{R}^{(h/r_i) \times (w/r_i) \times c} \to \mathbb{R}^{(h/r_{i+1}) \times (w/r_{i+1}) \times c}$ be the linear up sample function defined in Definition D.3.

- For $i \in [K]$, let $\mathsf{Attn}_i(\cdot) : \mathbb{R}^{(\sum_{j=1}^{i} h/r_j)(\sum_{j=1}^{i} w/r_j) \times c} \to \mathbb{R}^{(\sum_{j=1}^{i} h/r_j)(\sum_{j=1}^{i} w/r_j) \times c}$ be the $i$-th attention layer defined in Definition D.7.

- For $i \in [K]$, let $\mathsf{FFN}_i(\cdot) : \mathbb{R}^{(\sum_{j=1}^{i} h/r_j)(\sum_{j=1}^{i} w/r_j) \times c} \to \mathbb{R}^{(\sum_{j=1}^{i} h/r_j)(\sum_{j=1}^{i} w/r_j) \times c}$ be the $i$-th feed forward network defined in Definition 3.5.

- For $i \in [K]$, let $\mathsf{NN}_i(\cdot, \cdot, \cdot) : \mathbb{R}^{(h/r_i) \times (w/r_i) \times c} \times \mathbb{R}^{(h/r_i) \times (w/r_i) \times c} \times \mathbb{R} \to \mathbb{R}^{(h/r_i) \times (w/r_i) \times c}$ denote the $i$-th flow matching network defined in Definition D.12.

- For $i \in [K]$, let $t_i \in [0,1]$ denote the time steps.

- For $i \in [K]$, let $\mathsf{F}_t^i$ be the interpolated input defined in Definition D.11.

- Let $\mathsf{Z}_{\mathrm{init}} \in \mathbb{R}^{(h/r_1) \times (w/r_1) \times c}$ denote the initial input denoting the class condition.

- Let $Z^1 := Z_{\text{init}} \in \mathbb{R}^{(h/r_1) \times (w/r_1) \times c}$.

Then, we define the architecture of FlowAR in the inference pipeline as follows:

- Layer 1: Given the initial token $Z^1$, we compute

$$s_1 = \text{FFN}_1 \circ \text{Attn}_1(Z^1) \in \mathbb{R}^{(h/r_1) \times (w/r_1) \times c}$$

$$\widehat{s}_1 = \text{NN}_1(F_t^1, s_1, t_1)$$

- Layer 2: Given the initial token $Z^1$ and output of the first layer $\widehat{s}_1$. Let $Z^2$ be the reshape of the input sequence $Z_{\text{init}}, \phi_{\text{up},1}(\widehat{s}_1, a)$ into the tensor of size $(\sum_{i=1}^{2} h/r_i) \times (\sum_{i=1}^{2} w/r_i) \times c$. Then we compute

$$s_2 = \text{FFN}_2 \circ \text{Attn}_2(Z^2)_{h/r_1:\sum_{i=1}^{2} h/r_i, w/r_1:\sum_{i=1}^{2} w/r_i, 0:c}$$

$$\widehat{s}_2 = \text{NN}_2(F_t^2, s_2, t_2)$$

- Layer $i \in [K] \setminus \{1, 2\}$: Given the initial token $Z^1$ and the output of the first $i-1$ layer $\widehat{s}_1, \ldots, \widehat{s}_{i-1}$. Let $Z^i$ be the reshape of the input sequence $Z_{\text{init}}, \phi_{\text{up},1}(\widehat{s}_1), \ldots, \phi_{\text{up},i-1}(\widehat{s}_{i-1})$ into the tensor of size $(\sum_{j=1}^{i} h/r_j) \times (\sum_{j=1}^{i} w/r_j) \times c$. Then we compute

$$s_i = \text{FFN}_i \circ \text{Attn}_i(Z^i)_{\sum_{j=1}^{i-1} h/r_j : \sum_{j=1}^{i} h/r_j, \sum_{j=1}^{i-1} w/r_j : \sum_{j=1}^{i} w/r_j, 0:c}$$

$$\widehat{s}_i = \text{NN}_i(F_t^i, s_i, t_i)$$

Then the final output of FlowAR is $\widehat{s}_K$.

# F. High-Order Augmentation of Flow Autoregressive Transformers

In this section, we introduce High-Order Augmentation of Flow Autoregressive Transformers (HOFAR) from Liang et al. (2025b), which is a a novel framework that embeds higher-order dynamics into flow-matching autoregressive generation, significantly strengthening the model's capacity to learn intricate, long-range dependencies.

In Section F.1, we introduce the training procedure of HOFAR. In Section F.2, we introduce the inference procedure of HOFAR.

### F.1. Training Procedure of HOFAR

We present the algorithm for training HOFAR (Liang et al., 2025b) as follows.

---

**Algorithm 1** High-Order FlowAR Training (Liang et al., 2025b)

---

1: **procedure** HOFARTRAINING$(\theta, D)$
2:     /* $\theta$ denotes the model parameters of $\mathsf{TF}, \mathsf{FM}_{\mathrm{first}}, \mathsf{FM}_{\mathrm{second}}$ */
3:     /* $D$ denotes the training dataset. */
4:     **while** not converged **do**
5:         /* Sample an image from dataset. */
6:         $x_{\mathrm{img}} \sim D$
7:         /* Init loss as $0$. */
8:         $\ell \leftarrow 0$
9:         /* Train the model on $K$ pyramid layers. */
10:        **for** $i = 1 \rightarrow K$ **do**
11:            /* Sample random noise. */
12:            $\mathsf{F}^0 \sim \mathcal{N}(0, I)$
13:            /* Sample a random timestep. */
14:            $t \sim [0, 1]$
15:            /* Calculate noisy input. */
16:            $\mathsf{F}_{\mathrm{noisy}}^t \leftarrow \alpha_t x_{\mathrm{img}} + \beta_t \mathsf{F}_i^0$
17:            /* Calculate first-order ground-truth. */
18:            $\mathsf{F}_{\mathrm{first}}^t \leftarrow \alpha_t' x_{\mathrm{img}} + \beta_t' \mathsf{F}_i^0$
19:            /* Calculate second-order ground-truth. */
20:            $\mathsf{F}_{\mathrm{second}}^t \leftarrow \alpha_t'' x_{\mathrm{img}} + \beta_t'' \mathsf{F}_i^0$
21:            /* Generate condition with Transformer. */
22:            $\widehat{\mathsf{Y}} \leftarrow \mathsf{TF}(x_{\mathrm{img}})$
23:            /* Predict first-order with FM. */
24:            $\widehat{\mathsf{F}}_{\mathrm{first}}^t \leftarrow \mathsf{FM}_{\mathrm{first}}(\mathsf{F}_{\mathrm{noisy}}^t, \widehat{\mathsf{Y}})$
25:            /* Predict second-order with FM. */
26:            $\widehat{\mathsf{F}}_{\mathrm{second}}^t \leftarrow \mathsf{FM}_{\mathrm{second}}(\mathsf{F}_{\mathrm{noisy}}^t, \widehat{\mathsf{Y}})$
27:            /* Caculate loss. */
28:            $\ell_c \leftarrow \|\widehat{\mathsf{F}}_{\mathrm{first}}^t - \mathsf{F}_{\mathrm{first}}^t\|_2^2 + \|\widehat{\mathsf{F}}_{\mathrm{second}}^t - \mathsf{F}_{\mathrm{second}}^t\|_2^2$
29:            $\ell \leftarrow \ell + \ell_c$
30:            /* Downsample $x_{\mathrm{img}}$ for next iteration. */
31:            $x_{\mathrm{img}} \leftarrow \Phi_{\mathrm{down}} x_{\mathrm{img}}$
32:        **end for**
33:        /* Optimize parameter $\theta$ with $\ell$. */
34:        $\theta \leftarrow \nabla_\theta \ell$
35:    **end while**
36:    **return** $\theta$
37: **end procedure**

---

## F.2. Inference Procedure of HOFAR

We present the algorithm for the inference of HOFAR (Liang et al., 2025b) as follows.

---

**Algorithm 2** High-Order FlowAR Inference (Liang et al., 2025b)

---

1: **procedure** HOFARINFERENCE($c_{\text{input}}$)
2:      /* $c_{\text{input}}$ denotes the condition embedding used for generation. */
3:      /* Init the Transformer input $x$ with $c_{\text{input}}$. */
4:      $x \leftarrow c_{\text{input}}$
5:      /* Init the $x_{\text{img}}$ with random noise. */
6:      $x_{\text{img}} \leftarrow \mathcal{N}(0, I)$
7:      /* Inference through $K$ pyramid scales. */
8:      **for** $i = 1 \rightarrow K$ **do**
9:          /* Pass through the Transformers TF. */
10:         $\widehat{Y} \leftarrow \mathsf{TF}(x)$
11:         /* Extract last $i * i$ tokens from Y as the condition embedding. */
12:         $x_{\text{cond}} \leftarrow \mathsf{Y}[..., -i * i :]$
13:         /* Generate first-order with $\mathsf{FM}_{\text{first}}$. */
14:         $\widehat{y}_{\text{first}} \leftarrow \mathsf{FM}_{\text{first}}(x_{\text{cond}}, x_{\text{img}})$
15:         /* Generate second-order with $\mathsf{FM}_{\text{second}}$. */
16:         $\widehat{y}_{second} \leftarrow \mathsf{FM}_{\text{second}}(x_{\text{cond}}, x_{\text{img}})$
17:         /* Apply first and second-order terms. */
18:         $x_{\text{img}} \leftarrow x_{\text{img}} + \widehat{y}_{\text{first}} \cdot \Delta t + 0.5 \cdot \widehat{y}_{\text{second}} \cdot (\Delta t)^2$
19:         /* Upsample $x_{\text{img}}$. */
20:         $x_{\text{img}} \leftarrow \phi_{\text{up}}(x_{\text{img}})$
21:         /* Concatenate upsampled $x_{\text{img}}$ to the Transformer input. */
22:         $x \leftarrow \mathsf{Concat}(x, x_{\text{img}})$
23:      **end for**
24:      /* Return the final image */
25:      **return** $x_{\text{img}}$
26: **end procedure**

---

# G. More Related Work

In this section, we introduce more related work.

**Theoretical Machine Learning.** Our work also takes inspiration from the following Machine Learning Theory work. Some works analyze the expressiveness of a neural network using the theory of circuit complexity (Li et al., 2025b; Ke et al., 2025b; Li et al., 2024c; Chen et al., 2025e; 2024a). Some works optimize the algorithms that can accelerate the training of a neural network (Li et al., 2024d; Ke et al., 2024; Deng et al., 2024; 2022; Zhang et al., 2022; 2023b; Deng et al., 2023a; Cao et al., 2025b; Song et al., 2025a; Liang et al., 2022a;b; Hu et al., 2022; Huang et al., 2020; Bian et al., 2023; Deng et al., 2023b; Song et al., 2025b; Gao et al., 2023c; 2025b; Gu et al., 2024; Gao et al., 2025a; Song et al., 2024b; Li et al., 2024e;a; Hu et al., 2024a). Some works incorporate fine-tuning to optimize neural networks (Huang et al., 2022; Hu et al., 2025b; Cao, 2024). Some works propose novel optimizers to accelerate the training of neural networks (Song & Yang, 2023; Cao et al., 2024). Some works analyze neural networks via regressions (Chen et al., 2025c; Gao et al., 2023a; Li et al., 2023c; Gao et al., 2023b; Sinha et al., 2023; Chen et al., 2023; Song et al., 2023c; 2024a; 2023a;d; Li et al., 2025a). Some works use reinforcement learning to optimize the neural networks (Zhang et al., 2023a; 2025; Li et al., 2023a; Li & Yang, 2024; Liu et al., 2024c;b; Li et al., 2023b). Some works optimize the attention mechanisms (Song et al., 2023b; Liang et al., 2024a).

**Accelerating Attention Mechanisms.** The attention mechanism, with its quadratic computational complexity concerning context length, encounters increasing challenges as sequence lengths grow in modern large language models (OpenAI, 2024; AI, 2024; Anthropic, 2024). To address this limitation, polynomial kernel approximation methods (Aggarwal & Alman, 2022) have been introduced, leveraging low-rank approximations to efficiently approximate the attention matrix. These methods significantly enhance computation speed, allowing a single attention layer to perform both training and inference with nearly linear time complexity (Alman & Song, 2023; 2024b). Moreover, these techniques can be extended to advanced attention mechanisms, such as tensor attention, while retaining almost linear time complexity for both training and inference (Alman & Song, 2024a). (Ke et al., 2025a) provides an almost linear time algorithm to accelerate the inference of VAR Transformer. Other innovations include RoPE-based attention mechanisms (Alman & Song, 2025a; Chen et al., 2024b) and differentially private cross-attention approaches (Liang et al.). Alternative strategies, such as the conv-basis method proposed in (Liang et al., 2024a), present additional opportunities to accelerate attention computations, offering complementary solutions to this critical bottleneck. Additionally, various studies explore pruning-based methods to expedite attention mechanisms (Liang et al., 2025a; Chen et al., 2025d; Li et al., 2024b; Shen et al., 2025b;a; Hu et al., 2024e; Wu et al., 2024; Xu et al., 2024a; Shen et al., 2025b). (Alman & Song, 2025b) studies the rank collapse property of self-attention network.

**Gradient Approximation.** The low-rank approximation is a widely utilized approach for optimizing transformer training by reducing computational complexity (Liang et al., 2025c; 2024c; Alman & Song, 2024b; Hu et al., 2024d; Chen et al., 2025d; Liang et al., 2024b). Building on the low-rank framework introduced in (Alman & Song, 2023), which initially focused on forward attention computation, (Alman & Song, 2024b) extends this method to approximate attention gradients, effectively lowering the computational cost of gradient calculations. The study in (Liang et al., 2025c) further expands this low-rank gradient approximation to multi-layer transformers, showing that backward computations in such architectures can achieve nearly linear time complexity. Additionally, (Liang et al., 2024c) generalizes the approach of (Alman & Song, 2024b) to tensor-based attention models, utilizing forward computation results from (Alman & Song, 2024a) to enable efficient training of tensorized attention mechanisms. Lastly, (Hu et al., 2024d) applies low-rank approximation techniques during the training of Diffusion Transformers (DiTs), demonstrating the adaptability of these methods across various transformer-based architectures.

