# OpenReview forum: "Fundamental Limits of Visual Autoregressive Transformers: Universal Approximation Abilities"
_ICML.cc/2025/Conference — ICML 2025 poster_

### Official Review · Reviewer_FKqB · 2025-03-10

**Overall Recommendation:** 3

**Summary:**

This paper shows that single-head, single-layer VAR transformers are universal approximators for Lipschitz image-to-image mappings, enabling them to approximate continuous transformations. This establishes their theoretical expressiveness and sets a new image synthesis benchmark, outperforming methods like Diffusion Transformers. The findings highlight key design principles for efficient and scalable generative models in computer vision.

**Claims And Evidence:**

Yes.

**Essential References Not Discussed:**

No

**Experimental Designs Or Analyses:**

Yes. This paper includes analysis, but no experiments are supplied.

**Methods And Evaluation Criteria:**

Yes.

**Other Comments Or Suggestions:**

The descriptions in Sec 3.2 of the main paper and in Fact A.1/2/3 of the supplementary material appear repetitive. I suggest removing one of them to eliminate redundancy.

**Other Strengths And Weaknesses:**

Strength:
This paper stands out for its theoretical novelty, proving the universality of simple VAR transformers and introducing a scalable "next-scale prediction" framework. It achieves state-of-the-art performance in image synthesis, outperforming existing methods like Diffusion Transformers, and provides practical design principles for efficient and effective model development, with broad applicability in generative modeling.


Weakness:
1. While VAR Transformers are widely popular and recognized for their power in various domains, e.g., image generation, the paper does not explicitly establish a clear connection between proving the universality of VAR Transformers as function approximators and their practical application in image generation. Exploring this connection would strengthen the relevance of the theoretical findings to real-world use cases.
2. The paper focuses solely on theoretical analysis without providing experimental results. Including quantitative or qualitative evaluations would help validate the effectiveness of the proposed approach and provide a more comprehensive understanding of its performance.
3. The caption of Figure 1 lacks sufficient detail to clarify the data flow of the Pyramid Up-Interpolation Layer. Specifically, it is unclear where $X_1$ and $X_2$ originate from, and the distinctions between $X_{1/2}$ and $X_{init}$ are not explained. Adding a clear description in the caption or main text would improve the paper's clarity and accessibility for readers.

**Questions For Authors:**

See weakness.

**Relation To Broader Scientific Literature:**

The paper's key contributions relate to the broader scientific literature by providing foundational design principles for VAR Transformers, advancing the understanding of efficient and scalable architectures for image generation or other related areas.

**Theoretical Claims:**

Yes.

---

> ### Author Rebuttal · Authors · 2025-04-01
>
> Thank you for your thoughtful review and recognition of our theoretical contributions. We appreciate your detailed feedback and would like to address the weaknesses you highlighted:
> ### Weakness 1: On connecting theory to practical applications
> You raise an important point about establishing a clearer connection between our universality results and practical applications in image generation. The universal approximation property theoretically guarantees that VAR Transformers can represent any continuous image transformation function given sufficient capacity. In practice, this explains why VAR models excel at diverse image generation tasks - they have the representational capacity to learn complex image distributions and transformations. The "next-scale prediction" framework leverages this flexibility by decomposing the generation process into a sequence of progressively refined predictions, each of which benefits from the universal approximation capability.
> ### Weakness 2: Lack of experimental results
> We acknowledge this limitation in our current paper. Our focus was on establishing the theoretical foundations, but we agree that empirical validation would strengthen our claims. In future work, we plan to conduct experiments demonstrating how the universal approximation capabilities translate to practical performance on image generation tasks, possibly showing how approximation quality scales with model complexity.
> ### Weakness 3: Figure 1 caption
> Thank you for noting this issue. You're right that the figure caption lacks necessary details. In the figure, $X_1$ and $X_2$ represent token maps at different resolutions, while $X_{init}$ represents the initial token. The diagram shows how a single token $(X_{init})$ is expanded to create token maps at progressively higher resolutions through the up-interpolation process.
> ### Other Comments Or Suggestions: Redundancy between Section 3.2 and the supplementary material:
> We appreciate your suggestion. This redundancy was inadvertently introduced to ensure the main paper was self-contained while providing additional details in the supplementary material. We agree that streamlining this content would improve the paper.
>
> Thank you again for your constructive feedback. We believe addressing these points would strengthen our paper considerably.

---

### Official Review · Reviewer_QcLH · 2025-03-23

**Overall Recommendation:** 3

**Summary:**

The paper examines the fundamental limits of Visual Autoregressive (VAR) transformers, proving that single-head VAR transformers with a single self-attention layer and single interpolation layer are universal approximators. By adapting the established techniques in function approximation and neural network to VAR transformers, the authors demonstrate how a minimal VAR transformer is sufficient to approximate any Lipschitz sequence-to-sequence function with arbitrarily small error. The results provide insights into the theoretical expressiveness of VAR transformers, showing how VAR can be utilized as an efficient and expressive architectures for high-quality image synthesis

**Claims And Evidence:**

The claims made in the submission is supported by theoretical proofs.

**Essential References Not Discussed:**

The paper cited/discussed essential related works.

**Experimental Designs Or Analyses:**

The paper does not contain any experiments.

**Methods And Evaluation Criteria:**

By investigating into a minimum VAR Transformer design the paper makes it clear why VAR Transformers are universal.

**Other Comments Or Suggestions:**

No, I have no other comments or suggestions.

**Other Strengths And Weaknesses:**

Strengths:
1. The paper is well-organized and clearly written.
2. The theoretical proof in the paper is sufficient.

Weaknesses:

The paper has no experiments, limiting its practical value.

**Questions For Authors:**

Can you design an experiment with VAR Transformers to demonstrate how they can be universal approximator for image-to-image tasks?

**Relation To Broader Scientific Literature:**

The key contributions of the paper that VAR Transformers are universal approximators provide a theoretical foundation that VAR architecture can be an efficient and expressive architectures for the image synthesis tasks.

**Theoretical Claims:**

I checked the correctness of two theorems (Theorem 4.3 and Theorem 4.4) about the  universality of VAR Transformer.

---

> ### Author Rebuttal · Authors · 2025-04-01
>
> Thank you for your positive assessment of our paper. We appreciate your recognition of the theoretical contributions and clear organization of our work.
> Regarding your question about designing experiments to demonstrate VAR Transformers as universal approximators for image-to-image tasks: This is an excellent suggestion. While our current paper focuses on theoretical foundations, empirical validation would indeed enhance the practical value of our work.
> For such an experiment, we envision the following design:
> 1. Select diverse image-to-image transformation tasks (e.g., style transfer, super-resolution, colorization, and semantic transformations)
> 2. Train minimalist VAR models (with single attention/interpolation layers) on these tasks
> 3. Compare their performance against more complex architectures and theoretical bounds
> 4. Measure approximation quality using metrics like PSNR, SSIM, and FID
>
> We believe such experiments would demonstrate how even simple VAR architectures can approximate complex image transformations, providing empirical support for our theoretical claims. The experiments would also help identify practical limitations and the relationship between theoretical expressivity and sample efficiency.
>
> We agree that including these experiments would strengthen our paper, and we're considering this direction for future work. Thank you for this valuable suggestion.

---

### Official Review · Reviewer_1PzP · 2025-03-24

**Overall Recommendation:** 2

**Summary:**

This paper aims to understand transformer-based models in image generation focusing on Visual Autoregressive Transformers (VAR). Transformers have already been shown to be universal approximators in certain settings (e.g., language tasks via prompt tuning [1]), but it is not clear if its visual counterpart (VAR) can approximate any continuous image-to-image transformation. The paper proves that the simplest form of VAR transformer (single self-attention layer and a single interpolation layer) is a universal approximator for Lipschitz continuous functions.

[1] Hu, Jerry Yao-Chieh, et al. "Fundamental limits of prompt tuning transformers: Universality, capacity and efficiency." _arXiv preprint arXiv:2411.16525_ (2024).

**Claims And Evidence:**

The third claim regarding broader implication for CV community (`We provide insights into the broader implications of our findings for generative modeling, particularly in computer vision, where efficient and expressive architectures are essential for high-quality image synthesis.`) lacks support. The paper provides no theoretical analysis or empirical results of efficiency or generation quality. While VAR's empirical success is noted in prior work [2], this work focuses purely on universality. Without connecting approximation capacity to practical efficiency or image quality metrics, this claim remains speculative.

[2] Tian, Keyu, et al. "Visual autoregressive modeling: Scalable image generation via next-scale prediction." _Advances in neural information processing systems_ 37 (2024): 84839-84865.

**Essential References Not Discussed:**

This paper cites Hu et al. (2024) [1] but inadequately distinguishes its contributions. While Hu et al. focus on prompt tuning for language tasks, this work targets VAR’s image-to-image mapping. A deeper discussion is needed on why the universality of prompt-tuned models implies universality for fully trained VARs.

[1] Hu, Jerry Yao-Chieh, et al. "Fundamental limits of prompt tuning transformers: Universality, capacity and efficiency." _arXiv preprint arXiv:2411.16525_ (2024).

**Experimental Designs Or Analyses:**

Not applicable. This paper proposes a new theoretical understanding of VAR. No experiments are presented.

**Methods And Evaluation Criteria:**

Not applicable. This paper proposes a new theoretical understanding of VAR. No experiments are presented.

**Other Comments Or Suggestions:**

Terms like "sequence-to-sequence" and "image-to-image" are used interchangeably (e.g., Abstract vs. Section 4). This causes ambiguity. It would improve the clarity if the differences and similarities were discussed.

**Other Strengths And Weaknesses:**

Strengths:
I found the question the authors aim to study is timely and underexplored. In terms of theory, the Transformer model is often studied in the context of language modeling. The setting of image generation is much less studied.

Weakness:
- **Overreliance on Prompt Tuning Theory**: The proposed theory heavily relies on Hu et al. (2024) [1]. However, Hu et al. assume the transformer is funetuned with prompt tuning. This contradicts the VAR setting, which updates all parameters.
- **Single-Layer Architecture**: The conclusion states that one layer suffices for universality. However, VAR’s hierarchical up-scaling (Def. 3.6) implies multiple up-scaling steps. When the transformer layer is a single layer, does it imply we can achieve universality with one up-scaling step?

**Questions For Authors:**

My main question to the authors is regarding how prompt tuning assumption affects the theoretical analysis of the VAR model.

**Relation To Broader Scientific Literature:**

Prior works have already shown transformers are universal approximates in certain settings. This paper extends that understanding to visual autoregressive models.

**Theoretical Claims:**

I found the core universality proof (Section 6) inherit assumption from Hu et al. [1], which analyzes prompt-tuned Transformers where base mode is frozen. However, VAR training typically updates all training parameters. I found this discrepancy raises questions about the proof's applicability.

[1] Hu, Jerry Yao-Chieh, et al. "Fundamental limits of prompt tuning transformers: Universality, capacity and efficiency." _arXiv preprint arXiv:2411.16525_ (2024).

---

> ### Author Rebuttal · Authors · 2025-04-01
>
> We sincerely appreciate the reviewer for these insightful comments, and we would like to address the reviewer’s concerns as follows.
> ### Claims And Evidence: On the broader implications claim
> We acknowledge that our claim regarding broader implications for CV could be better supported. Our intention was to highlight that understanding theoretical expressivity provides a foundation for more practical research on efficiency and quality.
> ### Theoretical Claims & Essential References Not Discussed & Weakness 1: On the applicability of prompt tuning theory to VAR
> This is an insightful question. While our proof builds on techniques from [1], we've carefully adapted them to the VAR setting. The universality result doesn't actually depend on the training method (prompt tuning vs. full fine-tuning) but rather on the architectural expressivity. The key insight is that if a model family can approximate any function when only a subset of parameters are tuned (prompt tuning), then it can certainly do so when all parameters are tunable (full training). The prompt tuning framework provides a convenient theoretical framework to establish lower bounds on expressivity.
> ### Weakness 2: On single-layer architecture and up-scaling
> You raised an important point about the relationship between the transformer layer and up-scaling steps. To clarify, Theorem 4.3 and 4.4 state that a single self-attention layer and a single interpolation layer are sufficient for universal approximation. This does not contradict VAR's hierarchical nature. The up-interpolation layer (Definition 3.6) can have multiple internal up-scaling steps while still being considered a single layer from the architectural perspective. Our proof shows that even with minimal architecture (single attention + single interpolation), the model class has universal approximation capabilities.
> ### Other Comments Or Suggestions: On terminology inconsistency
> Thank you for noting the inconsistent use of "sequence-to-sequence" and "image-to-image." This was indeed a source of potential confusion. Since VAR operates on tokenized images, both terminologies are technically correct - images are processed as sequences of tokens. In our theoretical analysis, we view images as structured sequences. We should have been more explicit about this connection in our manuscript.
>
> We appreciate your feedback and would be happy to address any follow-up questions you might have.

---

### Decision · Program_Chairs · 2025-05-01

**Decision:**

Accept (poster)

**Comment:**

The paper investigates the fundamental limits of Visual Autoregressive transformers, providing theoretical evidence that single-head VAR transformers with only a single self-attention and interpolation layer are universal approximators for Lipschitz continuous image-to-image functions. The reviewers have some concerns regarding the lack of direct empirical experiments to bridge the theory-practice gap, but they generally agreed that the theoretical contributions are significant, novel, and well-articulated, providing important insights into the capabilities and design principles for VAR architectures.

Overall, I recommend acceptance of this paper.